# Modeling Combined Operation of Engine and Torque Converter for Improved Vehicle Powertrain's Complex Control

**Maksym Diachuk and Said M. Easa ***

Department of Civil Engineering, Ryerson University, 350 Victoria Street, Toronto, ON M5B2K3, Canada;
maksym.diachuk@ryerson.ca
* Correspondence: seasa@ryerson.ca

**Abstract:** This paper proposes an alternative model for describing the hydro-mechanical drive operation of the automatic transmissions. The study is aimed at preparing a reliable model that meets the requirements of sufficient informativeness and rapidity to, basically, be used as a model for optimized control. The study relevance is stipulated by the need for simple and precise models ensuring minimal computational costs in model predictive control (MPC) procedures. The paper proposes a method for coordinating the engine's control and operating modes, with the torque converter (TC) operating mode, based on the criteria of angular acceleration derivative (jerk), which fosters including the angular acceleration in the state vector for using the optimal control. The latter provides stronger prediction quality than using only the crankshaft angular speed criterion. This moment comprises a study novelty. Additionally, the proposed approach can be helpful in tasks of powertrain automation, autonomous vehicles' integrated control, elaboration of control algorithms, co-simulations, and real-time applications. The paper material is structured by the modeling stages, including mathematics and simulations, data preparation, testing and validation, virtual experiments, analysis of results, and conclusions. The essence of the problem, goals, and objectives are first presented, followed by the overview of main approaches in modeling the automatic transmission elements. The internal combustion engine (ICE), torque converter, drivetrain, tires, and translational dynamics mathematical models are determined and discussed in detail. The proposed approach convergence on decomposing the indicators of powertrain aggregates by derivatives is demonstrated. The considered method was simulated by using the data of the Audi A4 Quattro. The gear shifting control algorithm was described in detail, and a series of virtual tests for the composed model were carried out. The comparative analysis of the results for the conventional and advanced models of the automatic transmission's hydro-mechanical drive were performed. The differences of the model outputs were discussed. The approach advantages were noted, as well as the options for extending the proposed technique.

**Keywords:** automatic transmission; torque converter; powertrain dynamics; drivetrain modeling



## 1. Introduction

The emerging technologies of automating the vehicle control are aimed at excluding the influence of human factors on the vehicle handling character, its dynamics, and the operation of powertrain system units, as well as setting an objective of their optimal control, in coordination with other systems. Considering the increase of in-vehicle computer evaluation power, the use of approaches based on model predictive control (MPC) has become more popular and spread, contributing to replacing logical algorithms with optimal control. The MPC scheme requires two basic components: the plant (representing a real object, as completely as possible) and model (providing the state-space prognosis for a certain prediction horizon). Such models, as well as their elaboration, validation, approbation, testing, and tuning, represent a separate scientific interest to be used in the state-space form. On one hand, a model should be sufficiently informative; on the other

hand, it should exclude excessive mathematical complexity for unambiguous functioning, ensuring coding simplicity and high performance in real-time conditions.

Despite many simulation tools, such as Simulink/Simscape, the analytical approach (mathematical model) remains essential for understanding the interaction physics between the model components and validating the proposed method features, in general. Therefore, this research is dedicated to ensuring the practical elements of the powertrain model, for use in subsequent studies, in order to determine the optimal control. The main goal consists of considering the engine torque converter (TC) combination for modelling the space states of the system components and providing an adequate simulation of the automatic transmission functioning. The main task requires a mathematical description of the engine and torque converter interactions, which is complicated for several reasons: (1) the absence of a rigid connection between the impeller and turbine wheels and, consequently, the existence of an infinite number of possible solutions for the cooperative subsystem functioning; (2) the problem of the ICE's control accuracy, considering the TC's variable sensitivity, relative to changing its kinematic ratio; (3) the need for coordinating the gearbox shifting logic with the operating modes of both the ICE and the TC; and (4) the need to consider locking the TC with subsequent power transmitting over a torsional damper. In addition, to reflect the natural vehicle loading modes and operating circumstances, depending on external environmental conditions, a dynamic transmission model has to be composed, including a variable slope into dynamics equations.

A series of research in the field of modeling automatic transmission may be conditionally divided into three branches: simulation of automated powertrain components, studying the rational shifting algorithms, and optimizing the gearshift transients.

Usually, in engine control issues, the loading characteristic is assumed to be known and predictable, forming one external resistance curve, unlike the infinite quantity of them, in the case of a hydrodynamic loading. The state-space equations associated with achieving the system equilibrium by the engine control are usually composed, followed by linearizing both the driving and resisting torques. Thus, in general, the external load increment must be compensated by growing the cumulative engine torque caused by the torque adaptability, due to reducing the crankshaft speed and increasing the throttle position (fuel intake control). This approach works well when the relatively steady-state engine's operating conditions are maintained, but it may be insufficient at frequent and abrupt transition processes. Some papers on developing powertrain components are analyzed below.

De Araujo et al. [1] developed an automatic transmission model composed of three main subsystems, such as a torque convertor (TC) (including the forward and reverse flow modes), Lepelletier gearbox model, and gear shift schedule. The paper is focused on an improved TC model, in order to represent the transient and steady-state modes. These models are integrated into a vehicular dynamics and fuel consumption model, in order to be analyzed under a standard speed profile. Thus, our study's main idea consists of using a higher-order derivative for linking the states of powertrain aggregate dynamics, considering transients. Such an approach is expected to provide an improved model for use in optimal control problems.

Li et al. [2] evaluated the effect of torsional vibration appearing while engaging a clutch. Due to this, the shifting quality, accompanied by a noise of automatic transmissions, is decreased. An extended mathematical model of an automated powertrain was developed, considering nonlinear effects in a clutch and planetary gear. The dynamics of the stick-slip was described for the transition between the slipping clutches to the locked states. The gear backlash was considered for analyzing the planetary gear rattle noise. The authors outlined that the engagement process simulations showed the possibility of identifying the main factors affecting the noise generation.

Korendyasev et al. [3] simulated the gearshift process, with a shifting loop to be prevented. The transmission dynamic model includes the movable engine elements, a torque converter, and a gearbox input shaft (connected with a clutch and pinion). The model contains a control system generating shifting commands, based on the sensory

signal of output shaft rotational speed. In order to determine the coordinates and speeds of transmission elements before and after gear shifts, a series of relations was derived, followed by the numerical simulation of shift loops.

Zhang et al. [4] proposed a dynamic model of a six-speed automatic transmission with double-transition shifts. A planetary gearbox math model was derived, based on the Lagrange equation, and the dynamic analyses of two sets of clutches (ensuring double-transition shift) were carried out. The shift jerk and clutch energy loss were taken as describing a cost function to be used by the genetic algorithm in the optimization. The authors developed different strategies for the clutch overlapping time. It was remarked that the jerk effect may be reduced, as well as the energy loss and shift time.

Short et al. [5] emphasized the issues of modeling the longitudinal vehicle dynamics for hardware-in-the-loop (HIL) simulation. In the simulation, the car speed and position were determined using adaptive cruise control system, represented with several embedded microcontrollers. Different software architectures were utilized for distributed embedded systems. Millo et al. [6] considered the conventional powertrain model for issues of estimating the engine fuel consumption and pollutant emissions in different driving conditions.

Diachuk et al. [7] considered the full powertrain model, composed with Simulink/Simscape, to represent the vehicle dynamics features, including the steerability and roadability aspects. Hwang et al. [8] developed a complete powertrain to the analyze free and forced vibrations, as well as the natural frequencies of the torsional modes. The test measurements showed a good agreement with the virtual predictions.

In modern vehicles with automatic transmission, various algorithms are used to handle the powertrain logic, based on processing the data from several sensors and providing variable operating modes (sport, economy, etc.). The first automatic transmission control schemes proceeded from only two signals [9]: the crankshaft's angular speed sensor and load sensor on the gearbox output shaft. The load change caused a displacement of the central plunger distributing hydraulic flows to the high-pressure plungers that, in turn, controlled the planetary gear's hydraulic clutches. Some papers on organizing gearbox shifting are analyzed below.

Mishra et al. [10] studied the optimal pressure control for the friction clutches of two adjacent gears of an automatic transmission. First, the evolution of the system variables engaged in the upshift process were determined. The powertrain model was then composed considering the two clutches of the on-coming and off-going gears. Additionally, the model of the hydraulic system controlling the clutch was represented, including a command solenoid force. Using the model-based method and experimental results, the feedforward controller was generated, in order to accomplish all the phases of clutch engaging. The authors outlined the effectiveness of the controller proposed, as well as the convergence, robustness, and transient performance.

Mahmoud et al. [11] considered a mechanical transmission to be automated by using neuro-fuzzy control. All the components of an equivalent drivetrain were included into the translational vehicle dynamics model. The input parameters to be used by the controller for shift decision-making were the throttle position and vehicle wheel revolution. The trained shift control surface provides a robust control system for reaching the maximum performance. The authors asserted that the developed system can predict the gearshift performance for a vehicle model successfully.

In simulating the automatic transmission operation, some solutions for the gearshift scheme were based on a speed threshold, when the sets of upper and lower values for up- and downshifting were preliminarily established. An example of such a functioning model is discussed in [12]. An extended longitudinal dynamics model was developed for a passenger vehicle, which included the powertrain, braking control, controller unit, gearbox logic, and driver side control. As the primary input parameter, the reference model of the variable translational speed for a straight road section was set in a time interval of more than 2470 s. Based on the optimization process and considering many criteria,

such as power, ecologic factor, fuel efficiency, etc., the speed intervals for switching up and down were determined for each gear of the mechanical part. Thus, the resulting shift map contains two sets (upshift/downshift) of speed values, depending on the accelerator level and current gear. However, compared to the original scheme, the optimized map changes significantly, indicating an essential influence of external factors on the solution optimality. This example illustrates the possibility of using transmission models to solve practical problems, regarding rationalizing gearbox switching schemes. In this regard, the interest is to use the MPC for optimizing the gear switching, depending on upcoming road conditions (i.e., moving away from a static switching map to an online optimal one). Some papers on organizing shifting control are analyzed below.

Nezhadali and Eriksson [13] considered the optimal shifting control, based on an example of a nine-speed planetary gearbox. The working scheme of a powertrain model includes the diesel engine, torque converter, gearbox, final gear, flexible shaft drive, and wheel models. The matrix method was applied, in order to represent a system of powertrain dynamics and basic geometrical relation between planetary modules' components, as well. The optimal control of the gearshift was formulated as a compromise between the minimum switching time and integrated jerk (acceleration derivative) square. The authors used the nonlinear programming method to solve for the best control strategy for gearshift. According to the results received, the minimum jerk was reached in a series of simulation.

Newman et al. [14] aimed at the practical objective of rationalizing the gearshift schedule for a light-duty vehicle, using the full vehicle computer simulation and specific estimating software, based on MATLAB/Simulink, which can dynamically generate transmission logic from a set of user-defined parameters. Compared to initial, static, table-based shift logic, ALPHAshift software tool can tune the shift points, calibrating them for the same vehicle during testing. The basic principle of the tool algorithm consists of optimizing the fuel economy within the defined boundaries, which differ from pure optimization, emphasizing the fuel intake first. ALPHAshift includes cost, speed, performance, and setup parameters. The authors resumed that the ALPHAshift algorithm worked well, ensuring the automatic adjustment of custom parameters and tuning of shift points.

Tao et al. [15] described the math-based virtual vehicle environment system simulation based on General Motors Company's Road-to-Lab-to-Math (RLM), regarding its key components and application to transmission algorithm development. The transmission control algorithms reduce development times and costs, regarding physical prototypes. Wu et al. [16] considered the issues of simulating an automatic transmission of a tracked vehicle. The fuzzy controller is designed to organize shifting logic. The results of simulation showed the stable gearshift process without looping. Xi et al. [17] proposed a method for obtaining an optimized, dynamic, three-parameter gear shift algorithm of an eight-speed automatic transmission, based on vehicle acceleration characteristics, over MATLAB programming. The vehicle system integration model consisted of an engine, TC, gearbox, gear shift algorithm, and vehicle body dynamic system. The focus is in predicting the gear shift performance of a target vehicle.

Jeoung et al. [18] proposed an optimal gearshift strategy based on the greedy control method using the predicted velocity. A powertrain model was designed for forecasting the vehicle states after gear shifting with the predicted velocity. The strategy proposed was validated by simulating the urban driving cycle. Results show the capacity to increase the fuel efficiency, compared to the shift patterns of throttle and velocity. Meng and Jin [19] provided a new approach for solving the automatic gearbox strategy on sloping roads, based on a high-precision digital map. A set of driver-in-loop co-simulation tests were conducted, using a driving simulator equipped with a MATLAB/Simulink dynamics simulation platform. The novel intelligent shifting strategy ensures better driving performance on hilly roads. Sun et al. [20] evaluated a gearshift assistant mechanism for reducing the torque interruption and drivetrain jerk for traditional automated manual transmission by controlling presumed gearshift performance. Eckert et al. [21] developed an algorithm

for optimizing the gear shifting process modelled by co-simulation between the ADAMS multibody dynamics and Matlab/Simulink software.

Based on the review above, the powertrain dynamics constitutes a wide range of vehicle control tasks. The aforementioned authors have demonstrated various approaches and methods in achieving effective results. However, there is lack of attention devoted to the approaches that simultaneously provide both the best efficiency of torque converter and minimum fuel consumption, along with high performance indicators and a reduction of a gearshift number. Most of the control algorithms are based on ready-made or approximate automatic gearbox logic maps. Thus, this study, as part of a future series, sets the task of preparing, verifying, and testing the mathematical basis for the parts of the transmission model of an all-wheel drive vehicle with multiple control parameters. This work will serve as a stage for compiling a vehicle model to HIL simulation, followed by the development of schemes for optimizing both the operation of the powertrain and vehicle steering control for the prospect of autonomous handling.

The paper is organized as follows. Section 1 presents a literature review of the previous research. Section 2 considers composing the models of powertrain components, such as the engine, torque converter, friction clutch, general algorithm of automatic transmission control, tires, and a general model of vehicle translational dynamics. Two models are synthesized (conventional and advanced). Section 3 validates the decomposition approach using the derivative components of the engine, torque converter, and friction clutch models. Section 4 describes, in detail, the proposed Simulink model of the translational dynamics, including the transmission model. The stateflow model of the gearbox shift control is described, in detail, according to the algorithm proposed. The vehicle translational dynamics model is verified by comparing the simulation results of maximum performance mode with the source data; a series of tests were performed to demonstrate the automatic adaptation of models proposed to conditions with different combinations of the slope and fuel consumption variables. Section 5 draws conclusions about the applicability of the results in future studies.

## 2. System Modeling

### 2.1. Engine (E)

**Engine torque.** Assume that a spark-ignition engine (SIE) may be represented as a torque source with a constant inertia of rotating masses. However, the actual moment of inertia, due to changing positions of pistons, is a variable that is smoothed by a flywheel inertia. Based on the engine control method, an appropriate torque sufficient for stable operation in a stationary mode may be represented as follows:

$$T_e = T_e(\psi(t), \omega_e(t)) \tag{1}$$

where $\psi(t)$ is the accelerator (throttle) activation level, and $\psi \in [0, 1]$, $\omega_e(t)$ is the SIE's crankshaft angular speed.

The engine performance indicators may be represented by the speed characteristics, composing the so-called engine map, which is determined on a special stand in stationary conditions. A change of external load with a fixed fuel supply leads to decreasing the engine speed, forming a torque response curve. In practice, the most usable way implies proceeding from an external speed characteristic, corresponding to the maximum fuel supply and providing a torque curve, i.e., $T_{ext} = T_{ext}(\omega_e)$, which is generally described by an approximating or interpolating curve. With partial fuel supply, the curve changes in regard to the shape and range of operating revolutions, which may be reflected by the influence functions associated with the throttle position and engine shaft's current speed.

$$f(\psi) = \psi e^{k_\psi(1-\psi)} = \psi f_{\psi e}(\psi); \; f_\omega = f_\omega(\omega_e) = a\omega_e^3 + b\omega_e^2 + c\omega_e + d \tag{2}$$

where $k_\psi$ = intensity factor reflecting the power gain sensitivity settings, depending on the accelerator position, and the *a*, *b*, *c*, *d* = coefficients of the engine braking characteristic correspond to the null accelerator position. Then, SIE torque:

$$T_e = f_\psi T_{ext} + (1 - f_\psi) f_\omega \qquad (3)$$

As a modeling object, consider the Audi A4 Quattro's [22] 3.2 FSI engine map, obtained based on the external power characteristic [23] (Figure 1, red curve) and defined, with high precision, by a piecewise interval interpolation.

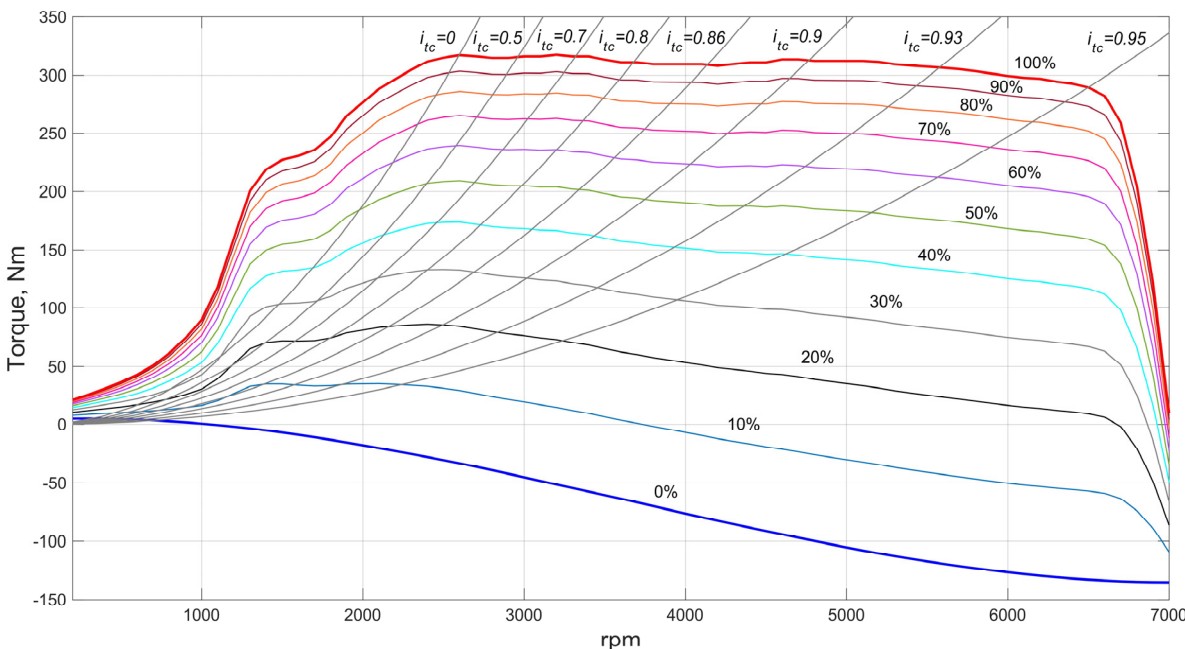

**Figure 1.** Speed–torque characteristic (map) of the 3.2 FSI engine for each 10% of fuel supply (throttle position) increment, combined with the TC's impeller loading characteristics.

**Torque derivative.** Since both the accelerator position $\psi$ and engine shaft speed $\omega_e$ are functions of time, and the rate of torque change is expressed as

$$\frac{d}{dt} T_e(\psi(t), \omega_e(t)) = \frac{\partial T_e}{\partial \psi} \frac{d\psi}{dt} + \frac{\partial T_e}{\partial \omega_e} \frac{d\omega_e}{dt} \qquad (4)$$

Denote $\varepsilon_e = d\omega_e/dt$ as the engine shaft's angular acceleration and $\omega_\psi = d\psi/dt$ as the angular speed of accelerator axis rotation. Such a variant, when the angular velocity $u_\psi = \omega_\psi$ is taken as the control signal, adequately agrees with the accelerator drive implementation by an electric motor when its shaft rotates with a constant angular speed, while changing the throttle position by a linear dependency within the discrete time step. This approach corresponds well with the control physics, unlike the case when direct signal $\psi$ is chosen. Its step-character will be equivalent to an instantaneous change of accelerator position, which does not consider transients and affects delays in the engine response, leading to its unstable operation, as well. Consequently, even the accelerator shaft angular rate ($u_\psi$) must be restricted for providing sustainable and smoothed regulation, avoiding spikes between adjacent states of engine shaft angular acceleration. Thus,

$$\frac{dT_e}{dt} = \frac{\partial T_e}{\partial \omega_e} \varepsilon_e + \frac{\partial T_e}{\partial \psi} u_\psi \qquad (5)$$

Using Equation (2), the corresponding components of engine torque changes by the frequency of shaft rotation, and the level of accelerator activation (link to the appendix) can be obtained as:

$$\frac{\partial T_e}{\partial \psi} = \frac{\partial f_\psi}{\partial \psi}(T_{ext} - f_\omega), \; \frac{\partial T_e}{\partial \omega_e} = f_\psi \frac{\partial T_{ext}}{\partial \omega_e} + (1 - f_\psi)\frac{\partial f_\omega}{\partial \omega_e} \tag{6}$$

where

$$\frac{\partial T_{ext}}{\partial \omega_e} = \frac{d}{d\omega_e}T_{ext}(\omega_e), \; \frac{\partial f_\psi}{\partial \psi} = (1 - \psi k_\psi)f_{\psi e}, \; \frac{\partial f_\omega}{\partial \omega_e} = 3a\omega_e^2 + 2b\omega_e + c \tag{7}$$

Since $T_{ext}$ depends only on the shaft rotation frequency ($\omega_e$), the full and partial derivatives are coincident, which value in a current point ($\omega_e$) may be obtained by differentiating piecewise polynomial functions representing the curve.

### 2.2. Torque Converter

The torque converter (TC) is a hydro-dynamic machine, where a turbine shaft torque is caused by the superposition of the active (impeller) and reactive (reactor) torque components. The transmission of Audi A4 Quattro includes the complex torque converter combining the advantages of both the TC and fluid coupling, possessing an option for forcible locking by a friction clutch (lock-up clutch) starting from the third gear speed. The impeller loading moment ($T_I$), while TC is operating as a hydro-dynamic machine, can be expressed as follows:

$$T_I = \rho D_a^5 \lambda_I \omega_I^2 \tag{8}$$

where $\rho$ = density of hydro-dynamic machine's working fluid, $D_a$ = TC active (maximum) diameter, $\lambda_I$ = impeller torque coefficient, and $\omega_I$ = impeller angular speed.

**TC characteristics.**

*Kinematic ratio.* Considering that $\omega_T$ is the turbine angular speed, yields:

$$i_{tc} = \omega_T / \omega_I \tag{9}$$

*Torque ratio.* Considering the moment $T_T$ on the turbine wheel:

$$k_{tc} = k_{tc}(i_{tc}) = T_T / T_I \tag{10}$$

Impeller torque coefficient:

$$\lambda_I = \lambda_I(i_{tc}) \tag{11}$$

The *efficiency* of the torque converter is defined as the ratio of the turbine power ($P_T$) to the impeller power ($P_I$):

$$\eta_{tc} = \frac{P_T}{P_I} = \frac{T_T \omega_T}{T_I \omega_I} = k_{tc} i_{tc} \tag{12}$$

Note that, despite the hydro-mechanical transmission advantages, the main drawback is the TC reduced efficiency at the modes with low kinematic ratio ($i_{tc}$) ($T_T$ is much higher than $T_I$) and when $k_{tc} = 1$. This leads to a fuel consumption increase of up to 7–10% on average, relative to an equivalent mechanical drivetrain. In this regard, the recommended TC's principal operational range of $i_{tc}$ must correspond to $\eta_{tc}$ above 80% [9].

Figure 2 depicts the TC's dimensionless characteristic adopted for modeling the cooperative work with the SI engine of the Audi A4 Quattro, where the impeller coefficient ($\lambda_I$) and torque ratio ($k_{tc}$) are presented as functions of $i_{tc}$ by the Lagrange polynomials of ***n*** and ***m*** orders, respectively:

$$\lambda_I = \sum_{k=0}^{n} p_k i_{tc}^k, \; k_{tc} = \sum_{k=0}^{m} q_k i_{tc}^k \tag{13}$$

where $p_k$ и $q_k$ = polynomial coefficients.

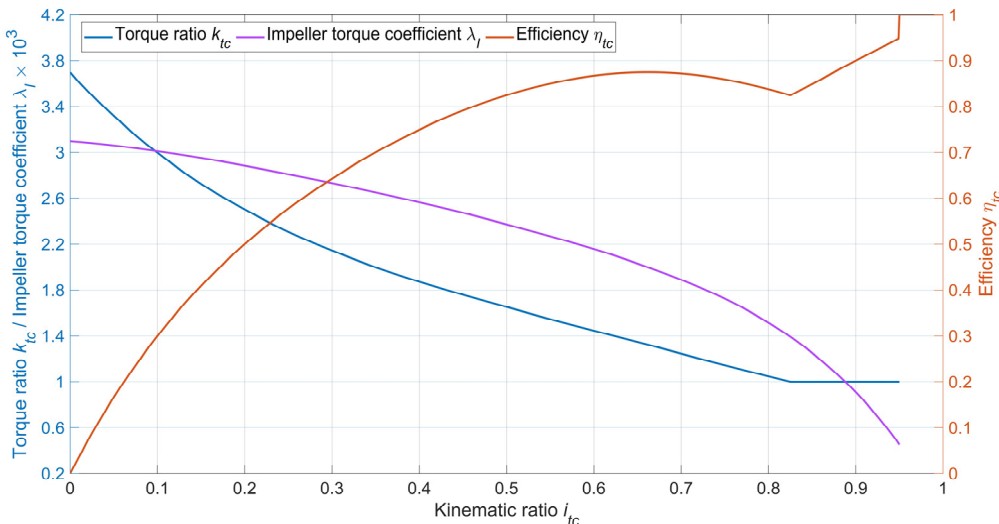

**Figure 2.** TC dimensionless characteristic.

Note that the TC characteristics are universal, due to the hydrodynamic machine similitude that allows for estimating the TC's geometric parameters for a newly designed hydro-mechanical drivetrains. This procedure is known as coordinating the ISE and TC characteristics by evaluating the active (maximum) diameter from Equation (8). The TC characteristic must correspond to the vehicle functionality, in order to ensure the power unit desirable operating modes, according to the target conditions of vehicle exploitation. Thus, in the case of the all-wheel drive and high-speed vehicle, the use of engine's both high-torque and -frequency operating performances would be the most expedient. Therefore, the TC must have the so-called direct transparency, which provides the impeller sensibility to the turbine loading changes. Thus, the engine shaft speed is reduced with the impeller load increase, resulting fewer piston strokes, but keeping the high-torque mode (Figure 1). On the other hand, there is a need for ensuring an option of transmitting the high-frequency power flow for using engine speed properties, which facilitates a wider control range by both the engine and gearbox for the power units' better adaptability to riding conditions. Since the Audi A4 Quattro TC diameter is already determined ($D_a = 276.2$ mm) by design [24], the coordination procedure can be carried out by clarifying the impeller torque coefficient $\lambda_{I0}$ at the "stop" mode ($\omega_I = 0$). Considering Figure 1 $T_I = 317$ N·m, $\rho = 860$ kg/m$^3$, and $\omega_I = 2\pi 2600/60 = 272$ rad/s, we obtain:

$$\lambda_{I0} = \frac{317}{860 \cdot 0.2762^5 \cdot 272^2} = 3.1 \cdot 10^{-3} \tag{14}$$

In calculating $\lambda_I$ for the set of $i_{tc}$, a series of parabolas can be plotted (Figure 1), reflecting the impeller load at a constant $i_{tc}$ value. The formed curvilinear mesh shows the whole variety of possible states of the ICE-TC combined operation.

**Impeller torque derivative.** The impeller and turbine angular speeds are variable while operating. However, their ratio $i_{tc}$ may remain constant. Thus, the impeller loading torque can be expressed as a time dependency:

$$T_I = T_I(\lambda_I(i_{tc}(t)), \omega_I(t)) \tag{15}$$

Time derivative:

$$\frac{d}{dt} T_I(\lambda_I(i_{tc}(t)), \omega_I(t)) = \frac{\partial T_I}{\partial \lambda_I} \frac{d\lambda_I}{dt} + \frac{\partial T_I}{\partial \omega_I} \frac{d\omega_I}{dt} = \frac{\partial T_I}{\partial \lambda_I} \frac{\partial \lambda_I}{\partial i_{tc}} \frac{di_{tc}}{dt} + \frac{\partial T_I}{\partial \omega_I} \frac{d\omega_I}{dt} \tag{16}$$

Denote:

$$\lambda'_{i_{tc}} = \frac{\partial \lambda_I}{\partial i_{tc}}, \quad \varepsilon_I = \frac{d\omega_I}{dt}, \quad \varepsilon_T = \frac{d\omega_T}{dt} \tag{17}$$

where $\omega_T$, $\varepsilon_T$ = turbine shaft angular speed and acceleration, correspondingly, and $\varepsilon_I$ = impeller shaft angular acceleration.

Consider the kinematic ratio $i_{tc}$ change by time:

$$\frac{di_{tc}}{dt} = \frac{d}{dt}\left(\frac{\omega_T}{\omega_I}\right) = \frac{1}{\omega_I^2}\left(\frac{d\omega_T}{dt}\omega_I - \omega_T\frac{d\omega_I}{dt}\right) = \frac{\varepsilon_T}{\omega_I} - \frac{\omega_T}{\omega_I}\frac{\varepsilon_I}{\omega_I} = \frac{1}{\omega_I}\left(\varepsilon_T - i_{tq}\varepsilon_I\right) \quad (18)$$

Consequently, it yields:

$$\frac{dT_I}{dt} = \frac{\partial T_I}{\partial\lambda_I}\frac{\lambda'_{i_{tc}}}{\omega_I}\left(\varepsilon_T - i_{tq}\varepsilon_I\right) + \frac{\partial T_I}{\partial\omega_I}\varepsilon_I = \left(\frac{\partial T_I}{\partial\omega_I} - \frac{\partial T_I}{\partial\lambda_I}\frac{\lambda'_{i_{tc}}}{\omega_I}i_{tq}\right)\varepsilon_I + \frac{\partial T_I}{\partial\lambda_I}\frac{\lambda'_{i_{tc}}}{\omega_I}\varepsilon_T \quad (19)$$

The components in the partial derivatives can be obtained based on Equation (8):

$$\frac{\partial T_I}{\partial\lambda_I} = \frac{\partial}{\partial\lambda_I}\left(\rho D_a^5\lambda_I\omega_I^2\right) = \rho D_a^5\omega_I^2, \frac{\partial T_I}{\partial\omega_I} = \frac{\partial}{\partial\omega_I}\left(\rho D_a^5\lambda_I\omega_I^2\right) = 2\rho D_a^5\lambda_I\omega_I \quad (20)$$

According to Equation (13):

$$\lambda'_{i_{tc}} = \frac{\partial\lambda_I}{\partial i_{tc}} = \frac{\partial}{\partial i_{tc}}\sum_{k=1}^{n}p_k i_{tc}^k = \sum_{k=1}^{n}kp_k i_{tc}^{k-1} \quad (21)$$

### 2.3. ICE-TC Combined Model

As noted in Figure 1, the cooperative work of the engine and torque converter imposes limitations on the engine operating modes within the outer parabolas ($i_{tc}$ equals 0 and 0.95); provided that there is no TC lock-up, the engine cannot use its entire map. Assume that the braking and propulsion functions are entirely separated between powertrain units, and no engine braking is used. That is, the TC transmits the power only in one direction. The engine shaft equation of motion is:

$$I_e\varepsilon_e = T_e - T_I \quad (22)$$

where $I_e$ is reduced to the engine shaft inertia of its moving masses with rigidly linked elements, including flywheel and TC impeller, and $\varepsilon_e = d\omega_e/dt$ is the engine shaft angular acceleration.

Equation (22) implies that $\varepsilon_e$ at least remains constant within an integration time step, which considers torque step changes during this period (caused by the transmission loading and engine control) unless the linearization approach is used. In this regard, consider the higher derivative:

$$\frac{d\varepsilon_e}{dt} = \frac{1}{I_e}\left(\frac{dT_e}{dt} - \frac{dT_I}{dt}\right) \quad (23)$$

Since there is no speed reducer between the engine shaft and TC impeller, then:

$$\omega_I = \omega_e, \varepsilon_I = \varepsilon_e \quad (24)$$

Substituting expressions of Equations (5) and (19) in Equation (23), obtain a differential relation, where $\varepsilon_e$ is already represented at least by a piecewise linear function:

$$\frac{d\varepsilon_e}{dt} = \frac{1}{I_e}\left(\frac{\partial T_e}{\partial\psi}u_\psi + \frac{\partial T_e}{\partial\omega_e}\varepsilon_e - \left(\left(\frac{\partial T_I}{\partial\omega_e} - \frac{\partial T_I}{\partial\lambda_I}\frac{\lambda'_{i_{tc}}}{\omega_e}i_{tq}\right)\varepsilon_e + \frac{\partial T_I}{\partial\lambda_I}\frac{\lambda'_{i_{tc}}}{\omega_e}\varepsilon_T\right)\right) =$$
$$= \frac{1}{I_e}\left(\left(\frac{\partial T_e}{\partial\omega_e} - \frac{\partial T_I}{\partial\omega_e} + \frac{\partial T_I}{\partial\lambda_I}\frac{\lambda'_{i_{tc}}}{\omega_e}i_{tq}\right)\varepsilon_e - \frac{\partial T_I}{\partial\lambda_I}\frac{\lambda'_{i_{tc}}}{\omega_e}\varepsilon_T + \frac{\partial T_e}{\partial\psi}u_\psi\right) \quad (25)$$

It can be noted that, with this approach, $\varepsilon_e$ is either known as a result of preceding integrating the Equation (25) or can be determined using Equation (22). Both ways are

the same. In its turn, $\varepsilon_T$ can be found using the turbine power balance using a vehicle drivetrain model.

### 2.4. Gearbox Logic (G)

Within this study, the central importance focuses on modeling the interaction between the EC engine and TC in a wide range of power and speed modes. In this regard, the powertrain control strategy should be based on sustaining the specific TC's sliding modes. On the one hand, it is logical to use high torque transformation regimes that require lower transmission ratios. On the other hand, low sliding turbine modes may be used, but the needed transmission ratios are increased. However, the best simple criterion for determining a gear switching moment is the current TC's efficiency [24]. Due to the TC's converting properties, the engine can operate with any gear, even if the load on the turbine shaft leads to its complete stop. Considering performance and fuel economy, the TC's sliding modes providing an efficiency higher than 0.8 [25] are expedient (Figure 3a), which corresponds to the desirable range of the kinematic ratio $[i_{min}, i_{max}]$. In this regard, using Figure 2, the recommended range should be $0.47 \leq i_{tc} \leq 0.95$. The lower limit requirement is unnecessary for the first gear, as it must ensure a vehicle's minimum stable speed, including slopes. Thus, the moment $i_{tc} \leq 0.47$ can be considered the trigger for downshifting, and $i_{tc} \geq 0.95$ can be considered the trigger for upshifting.

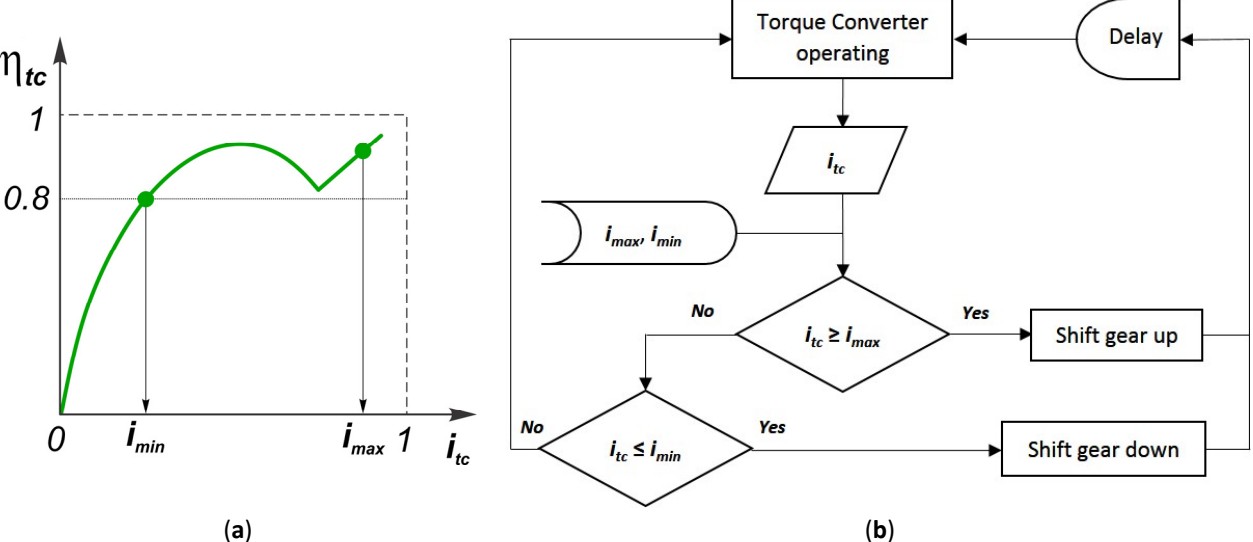

**(a)**                  **(b)**

**Figure 3.** Automatic transmission control strategy: (**a**) determining the efficiency zone greater than 0.8; (**b**) simple gear shifting algorithm.

A separate point regards the neutral and motion starting. Thus, for the first and reverse gears, the lower limit $i_{tc} = 0$ is allowed. In this research, the TC forcible lock-up via using a friction clutch and torsional vibration damper is considered only for the highest gear.

### 2.5. Drivetrain Model (D-F-W)

Since this study focuses on modeling the interaction and cooperative work of the ICE and TC, the Audi A4 Quattro's powertrain may be schematized in a simplistic view (Figure 4). The SI engine (**E**) provides the propulsion torque $T_e$ on the crankshaft rigidly connected to the torque converter's (**TC**) impeller. The total inertia of all the elements that transmit the engine torque to the impeller blades is reduced to the crankshaft and denoted as $I_e$. The TC turbine is connected to the automatic gearbox input shaft, which stipulates their total moment of inertia ($I_T$). The impeller and turbine may interact directly, being mechanically connected by locking the TC's friction clutch. In a planetary gearbox, the

torque is converted in several stages, depending on a current gear (*j*) with a total ratio, i.e., $i_{g(j)}$. In this case, the inertia of the rotating masses reduced to the gearbox output shaft will also be variable, i.e., $I_{g(j)}$. The gearbox output shaft drives the inter-axle differential carrier (**D**) by a gear train with a ratio of $i_d$. Assume that the front and rear axles' drive shafts rotate at the same angular speeds synchronously with the inter-axle differential carrier. Then, all the specified masses may be reduced to the inertia $2I_d$. Each axle is driven through an inter-wheel differential, in which the carrier is jointed with a final gear (**F**) wheel, providing a ratio $i_f$. The influence of rotating masses associated with the wheels' drive (**W**), including the wheels themselves, is related to the inertia $4I_w$. Based on the assumption regarding the equal distribution of vertical reactions between the vehicle wheels and, consequently, traction forces, the conditional wheel is loaded with the external resistance torque $4M_w$.

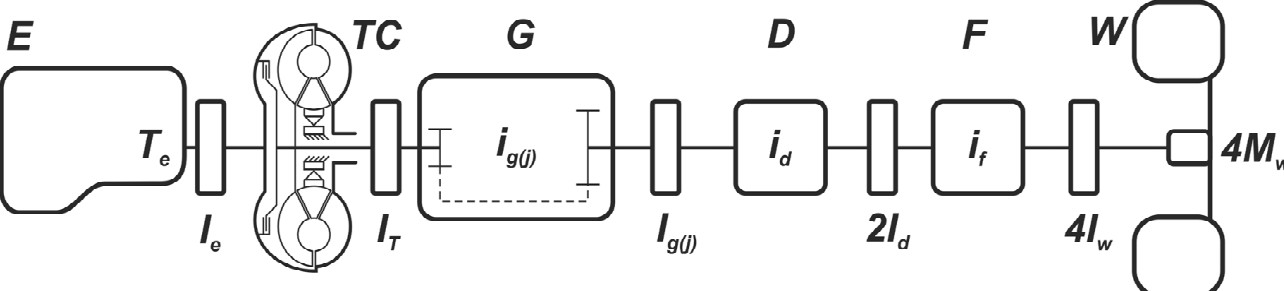

**Figure 4.** Scheme of conditional structure for all-wheel drive automatic transmission: **E**—engine, **TC**—torque converter, **G**—gearbox, **D**—drive of inter-axle differential, **F**—final gear, and **W**—wheels.

The composed dynamic equations for the drivetrain model are shown in Figure 4. Assume that the power flow is transmitted through the drive elements only in one direction, from the engine to wheels. That is, the option of braking by the engine is not supported, which eliminates the need for reverse efficiency. Dynamic balance of a wheel axle:

$$I_w \varepsilon_w = T_w/2 - G_w - B_w - L_w - M_w \tag{26}$$

where $I_w$ = total moment of inertia of a wheel and its drive rotating masses, $\varepsilon_w$ = wheel angular acceleration, $T_w/2$ = driving torque equal to a half of the torque on the carrier of inter-wheel symmetric differential, $G_w$ = moment of losses in gearing between the semi-axle driving pinion and satellites, $B_w$ = moment of losses in the wheel drive bearings and joints, $L_w = l_w \omega_w$ = moment of hydraulic (viscous) losses, and $M_w$ = external loading moment.

If the absence of driving torque results in nearly no gearing loss, then the inertial torque ought to partially compensate for the bearing losses associated with rolling resistance. Thus, using Equation (26), the balance of an axle can be represented in the following form:

$$2I_w \varepsilon_w \eta_{wB} = T_w \eta_{wG} \eta_{wB} - 2L_w - 2M_w \tag{27}$$

where $\eta_{wG}$ = the gearing efficiency of driving pinion and satellites, and $\eta_{wB}$ = the efficiency of joints and bearings.

The external moment resisting the wheel drive, excluding the braking torque, is represented as:

$$M_w = R_{wx} r_{wd} + M_{wy} \tag{28}$$

where $r_{wd}$ = wheel dynamic radius, $R_{wx}$ = longitudinal reaction of tire-road interaction, and $M_{wy}$ = pure rolling resistance moment.

The relations that tie the input and output kinematic parameters of the final gear (**F**) are given by:

$$i_f = \frac{\omega_d}{\omega_w}, \quad \varepsilon_w = \frac{\varepsilon_d}{i_f} \tag{29}$$

где $\omega_d$, $\omega_w$ = the angular speeds of the final gear's input and output shafts, respectively, and $\varepsilon_d$, $\varepsilon_w$ = the angular accelerations, correspondingly.

Substituting Equation (29) in Equation (27), the following expression for the driving torque of one axle's wheels may be derived:

$$T_w = \frac{2I_w}{\eta_{wG}i_f}\varepsilon_d + \frac{2L_w}{\eta_{wG}\eta_{wB}} + \frac{2M_w}{\eta_{wG}\eta_{wB}} \tag{30}$$

Consider the dynamic equilibrium of the final gear driven by the inter-axle differential output shaft, provided that the torque is conditionally divided in half between axles:

$$I_d\varepsilon_d = T_d/2 - G_d - B_d - L_d - T_w/i_f \tag{31}$$

$T_d/2$ = driving torque equal to a half of the torque on the carrier of inter-axle symmetrical differential, $G_d$ = moment of losses in gearing between the satellites and output shaft pinion of inter-axle differential, $B_d$ = moment of losses in the bearings and joints of final gear drive, and $L_d$ = moment of hydraulic (viscous) losses in final gear drive.

Similar to Equation (27):

$$\frac{2I_d}{\eta_{dG}}\varepsilon_d = T_d - \frac{2L_d}{\eta_{dG}\eta_{dB}} - \frac{2T_w}{i_f\eta_{dG}\eta_{dB}} \tag{32}$$

where $\eta_{dG}$ = gearing efficiency between the inter-axle differential satellites and its output shaft pinion and $\eta_{dB}$ = efficiency of bearings and joints in the final gear drive.

The kinematic parameters at the gearbox (G) output shaft and the carrier of inter-axle differential (D) are tied by relations:

$$i_d = \frac{\omega_{g(j)}}{\omega_d}, \quad \varepsilon_d = \frac{\varepsilon_{g(j)}}{i_d} \tag{33}$$

where $\omega_{g(j)}, \varepsilon_{g(j)}$ = the angular speed and acceleration of gearbox output shaft on a *j*th gear speed, respectively.

Substituting Equation (30) in Equation (32), the following expression for the inter-axle differential driving torque may be derived:

$$T_d = \left(\frac{4I_w}{i_f^2\eta_{wG}} + \frac{2I_d}{\eta_{dG}}\right)\frac{\varepsilon_g}{i_d} + \frac{2L_d}{\eta_{dG}\eta_{dB}} + \frac{1}{i_f\eta_{dG}\eta_{dB}}\left(\frac{4L_w}{\eta_{wG}\eta_{wB}} + \frac{4M_w}{\eta_{wG}\eta_{wB}}\right) \tag{34}$$

Consider the dynamic balance of the automatic gearbox output shaft in the *j*th gear speed by analogy with Equation (26)

$$I_{g(j)}\varepsilon_{g(j)} = M_Ti_{g(j)} - G_{g(j)} - B_{g(j)} - L_{g(j)} - T_d/i_d \tag{35}$$

where $M_T$ = driving torque on the automatic gearbox (G) input shaft, $G_{g(j)}$ = moment of losses in gearing of the automatic gearbox (G) at *j*th speed reduced to the output shaft, $B_{g(j)}$ = moment of losses of rolling resistance in bearings at *j*th speed of the automatic gearbox reduced to the output shaft, and $L_{g(j)}$ = moment of viscous losses in the automatic gearbox at *j*th speed reduced to the output shaft.

By the analogy with Equation (26), transform the Equation (35):

$$I_{g(j)}\varepsilon_{g(j)}\eta_{gB(j)} = M_Ti_{g(j)}\eta_{gG(j)}\eta_{gB(j)} - L_{g(j)} - \frac{T_d}{i_d} \tag{36}$$

where $\eta_{gG(j)}$, $\eta_{gB(j)}$ = gearing and bearing efficiencies of the automatic gearbox (G) at *j*th speed, respectively.

The relations that tie the kinematic parameters at the input and output gearbox shafts are given by:

$$i_{g(j)} = \frac{\omega_T}{\omega_{g(j)}}, \quad \varepsilon_{g(j)} = \frac{\varepsilon_T}{i_{g(j)}} \tag{37}$$

Substituting Equation (37) in Equation (36), the following expression for the automatic gearbox power balance can be obtained:

$$\frac{I_{g(j)}}{i_{g(j)}^2 \eta_{gG(j)}} \varepsilon_T = M_T - \frac{L_{g(j)}}{\eta_{gG(j)} \eta_{gB(j)} i_{g(j)}} - \frac{1}{\eta_{gG(j)} \eta_{gB(j)} i_{g(j)}} \frac{T_d}{i_d} \tag{38}$$

Substituting Equation (34) in Equation (38), the expression for the turbine shaft loading moment yields:

$$M_T = \frac{I_{g(j)}}{i_{g(j)}^2 \eta_{gG(j)}} \varepsilon_T + \frac{L_{g(j)}}{\eta_{gG(j)} \eta_{gB(j)} i_{g(j)}} + \frac{1}{\eta_{gG(j)} \eta_{gB(j)} i_{g(j)}} \frac{T_d}{i_d} = \frac{I_{g(j)}}{i_{g(j)}^2 \eta_{gG(j)}} \varepsilon_T + \frac{L_{g(j)}}{\eta_{gG(j)} \eta_{gB(j)} i_{g(j)}} +$$

$$+ \frac{1}{\eta_{gG(j)} \eta_{gB(j)} i_{g(j)}} \left( \left( \frac{4I_w}{i_f^2 \eta_{wG}} + \frac{2I_d}{\eta_{dG}} \right) \frac{\varepsilon_{g(j)}}{i_d^2} + \frac{2L_d}{i_d \eta_{dG} \eta_{dB}} + \frac{1}{i_d i_f \eta_{dG} \eta_{dB}} \left( \frac{4L_w}{\eta_{wG} \eta_{wB}} + \frac{4M_w}{\eta_{wG} \eta_{wB}} \right) \right) = \tag{39}$$

$$= \left( I_{g(j)} + \left( \frac{2I_d}{\eta_{dG}} + \frac{4I_w}{i_f^2 \eta_{wG}} \right) \frac{1}{i_d^2 \eta_{gB(j)}} \right) \frac{\varepsilon_T}{i_{g(j)}^2 \eta_{gG(j)}} + \frac{1}{i_{g(j)} \eta_{gG(j)} \eta_{gB(j)}} \left( L_{g(j)} + \frac{1}{i_d \eta_{dG} \eta_{dB}} \left( 2L_d + \frac{4L_w + 4M_w}{i_f \eta_{wG} \eta_{wB}} \right) \right)$$

Consider the turbine shaft dynamics balance by analogy with Equation (26):

$$I_T \varepsilon_T = T_T - B_T - L_T - M_T \tag{40}$$

where $T_T$—driving torque on turbine (TC) blades, $B_T$—moment of rolling losses in turbine shaft bearings, and $L_T$—moment of viscous losses in the turbine shaft drive.

Either introducing $\eta_{TB}$ = the efficiency of turbine shaft bearing, or Equation (40) will be rewritten in the form:

$$I_T \varepsilon_T \eta_{TB} = T_T \eta_{TB} - L_T - M_T \tag{41}$$

Substituting $M_T$ from Equation (39) in Equation (41), it is possible to link all of the drivetrain parameters:

$$I_T \varepsilon_T \eta_{TB} = T_T \eta_{TB} - L_T - \left( I_{g(j)} + \left( \frac{4I_w}{i_f^2 \eta_{wG}} + \frac{2I_d}{\eta_{dG}} \right) \frac{1}{i_d^2 \eta_{gB(j)}} \right) \frac{1}{i_{g(j)}^2 \eta_{gG(j)}} \varepsilon_T -$$

$$- \frac{1}{i_{g(j)} \eta_{gG(j)} \eta_{gB(j)}} \left( L_{g(j)} + \frac{1}{i_d \eta_{dG} \eta_{dB}} \left( 2L_d + \frac{4L_w + 4M_w}{i_f \eta_{wG} \eta_{wB}} \right) \right) \tag{42}$$

Collecting the inertial members on the left, and the others on the right, obtain:

$$\left( I_T + \left( I_{g(j)} + \left( \frac{4I_w}{i_f^2 \eta_{wG}} + \frac{2I_d}{\eta_{dG}} \right) \frac{1}{i_d^2 \eta_{gB(j)}} \right) \frac{1}{i_{g(j)}^2 \eta_{gG(j)} \eta_{TB}} \right) \varepsilon_T = T_T - \frac{L_T}{\eta_{TB}} -$$

$$- \frac{1}{i_{g(j)} \eta_{TB} \eta_{gG(j)} \eta_{gB(j)}} \left( L_{g(j)} + \frac{1}{i_d \eta_{dG} \eta_{dB}} \left( 2L_d + \frac{4L_w + 4M_w}{i_f \eta_{wG} \eta_{wB}} \right) \right) \tag{43}$$

Denote the drivetrain moment of inertia reduced to the turbine shaft:

$$I_{tr} = I_T + \left( I_{g(j)} + \left( \frac{2I_d}{\eta_{dG}} + \frac{4I_w}{i_f^2 \eta_{wG}} \right) \frac{1}{i_d^2 \eta_{gB(j)}} \right) \frac{1}{i_{g(j)}^2 \eta_{gG(j)} \eta_{TB}} \tag{44}$$

Moment of external and internal resistances reduced to the turbine shaft:

$$M_{tr} = \frac{1}{\eta_{TB}} \left( L_T + \frac{1}{i_{g(j)} \eta_{gG(j)} \eta_{gB(j)}} \left( L_{g(j)} + \frac{1}{i_d \eta_{dG} \eta_{dB}} \left( 2L_d + 4 \frac{L_w + M_w}{i_f \eta_{wG} \eta_{wB}} \right) \right) \right) \tag{45}$$

The viscous resistance moments tied to the transmission parts' angular velocities may be obtained as a vector. The rotational speeds of drivetrain shafts are expressed through the current $\omega_e$, $i_{tc}$, $i_{g(j)}$:

$$\boldsymbol{\omega} = \begin{pmatrix} \omega_T \\ \omega_{g(j)} \\ \omega_d \\ \omega_w \end{pmatrix} = \begin{pmatrix} 1 \\ 1/i_{g(j)} \\ 1/\left(i_d i_{g(j)}\right) \\ 1/\left(i_f i_d i_{g(j)}\right) \end{pmatrix} \omega_e i_{tc} \tag{46}$$

Then:

$$\boldsymbol{L} = \begin{pmatrix} L_T \\ L_{g(j)} \\ L_d \\ L_w \end{pmatrix} = \begin{pmatrix} l_T \omega_T \\ l_{g(j)} \omega_{g(j)} \\ l_d \omega_d \\ l_w \omega_w \end{pmatrix}, \boldsymbol{l} = \begin{pmatrix} l_T \\ l_{g(j)} \\ l_d \\ l_w \end{pmatrix}, \boldsymbol{L} = diag(\boldsymbol{l})\boldsymbol{\omega} \tag{47}$$

Then, the turbine angular acceleration may be expressed through the drivetrain dynamics:

$$\varepsilon_T = \frac{1}{I_{tr}}(T_T - M_{tr}) \tag{48}$$

Thus, substituting Equation (48) in Equation (26), the main equation of powertrain state may be represented as follows:

$$\frac{d\varepsilon_e}{dt} = \frac{1}{I_e}\left(\left(\frac{\partial T_e}{\partial \omega_e} - \frac{\partial T_I}{\partial \omega_e} + \frac{\partial T_I}{\partial \lambda_I}\frac{\lambda'_{i_{tc}}}{\omega_e}i_{tq}\right)\varepsilon_e - \frac{\partial T_I}{\partial \lambda_I}\frac{\lambda'_{i_{tc}}}{\omega_e}\frac{1}{I_{tr}}(T_T - M_{tr}) + \frac{\partial T_e}{\partial \psi}u_\psi\right) \tag{49}$$

*2.6. TC Lock-Up*

As seen in Figure 1, a significant part of the engine map is not covered by the TC operating modes, which leads to the underuse of the engine speed properties, when the TC converting capability is almost absent (mode $k_{tc} = 1$) and hydrodynamic flow only smooths the engine crankshaft load. In this case, the forcible lock-up of torque converter by a friction clutch is used, and further interaction between the engine shaft and automatic gearbox input shaft (turbine shaft) is carried out through a torsional vibration damper (TVD). In general, the clutch engagement is preceded by a slipping process; however, the difference of the turbine and impeller shafts' angular speeds is, in this case, less than 5%. Within the framework of this project objective, it can be assumed that the clutch engagement occurs almost instantly. Thus, the driving torque is directly passed by a TVD, consisting of independent elastic and damping parts and allowing a certain relative angular displacement between the damper disc and its hub. Then, the moment in the damper can be expressed as:

$$T_D = T_D(\Delta\phi, \Delta\omega) = T_{D\phi}(\Delta\phi) + T_{D\omega}(\Delta\omega) \tag{50}$$

where $\Delta\phi = \phi_e - \phi_T$—difference of the engine and turbine shafts' rotation angles, $\Delta\omega = \omega_e - \omega_T$—difference of the engine and turbine shafts' angular speeds, and $T_{D\phi}$, $T_{D\omega}$—elastic and absorbing components of the TVD moment.

Time derivative yields:

$$\frac{d}{dt}T_D(\Delta\phi, \Delta\omega) = \frac{\partial T_D}{\partial \Delta\phi}\frac{d\Delta\phi}{dt} + \frac{\partial T_D}{\partial \Delta\omega}\frac{d\Delta\omega}{dt} = \frac{\partial T_{D\phi}}{\partial \Delta\phi}(\omega_e - \omega_T) + \frac{\partial T_{D\omega}}{\partial \Delta\omega}(\varepsilon_e - \varepsilon_T) \tag{51}$$

Then, considering that the kinematic relation (9) remains relevant, Equation (23) takes the form:

$$\frac{d\varepsilon_e}{dt} = \frac{1}{I_e}\left(\frac{\partial T_e}{\partial \psi}u_\psi + \frac{\partial T_e}{\partial \omega_e}\varepsilon_e - \left(\frac{\partial T_{D\phi}}{\partial \Delta\phi}(\omega_e - \omega_T) + \frac{\partial T_{D\omega}}{\partial \Delta\omega}(\varepsilon_e - \varepsilon_T)\right)\right) =$$
$$= \frac{1}{I_e}\left(\left(\frac{\partial T_e}{\partial \omega_e} - \frac{\partial T_{D\omega}}{\partial \Delta\omega}\right)\varepsilon_e - \frac{\partial T_{D\phi}}{\partial \Delta\phi}(1 - i_{tc})\omega_e + \frac{\partial T_{D\omega}}{\partial \Delta\omega}\varepsilon_T + \frac{\partial T_e}{\partial \psi}u_\psi\right) \tag{52}$$

where the turbine shaft angular acceleration at $T_T = T_D$.

$$\varepsilon_T = \frac{1}{I_{tr}}(T_D - M_{tr}) \tag{53}$$

However, unlike the case of TC operating in converting mode, the mechanical connection of shafts causes the introduction of an additional state parameter—the angular damper displacement (deformation):

$$\frac{d\Delta\phi}{dt} = \frac{d}{dt}(\phi_e - \phi_T) = \omega_e - \omega_T = (1 - i_{tc})\omega_e \tag{54}$$

The TVD's elastic component design must match its operating conditions. Since engine braking in an automatic transmission is not allowed, there is no need for neither the characteristic's physical symmetry nor for the excessive rigidity in the negative deformation phase. Thus, the damper design may consist of a two-stage system of elastic elements changing the angular stiffness at the positive deformation phase (the engine shaft outstrips the turbine shaft), as well as a single-stage system at the negative phase (the turbine shaft outpaces the engine shaft). Thus, the piecewise linear functions can be used for describing the damper elastic torque:

$$T_{D\phi}(\Delta\phi) = \left(c_\phi\Delta\phi + b_\phi\right)^T(E_-(\Delta\phi - \Delta\phi_l) - E_-(\Delta\phi - \Delta\phi_r)) \tag{55}$$

where $c_\phi$ = angular stiffness vector, $b_\phi$—vector of linear functions' free coefficients, $E_-(\Delta)$ = step function equal to 1 if $\Delta \geq 0$, and $\Delta\phi_l$, $\Delta\phi_r$ = vectors of left- and right-side boundaries of elastic characteristic's linear sections, respectively.

Denote:

$$c_\phi = \begin{pmatrix} c_{\phi n} \\ c_{\phi c} \\ c_{\phi p} \\ c_{\phi e} \end{pmatrix}, \quad b_\phi = \begin{pmatrix} b_{\phi n} \\ b_{\phi c} \\ b_{\phi p} \\ b_{\phi e} \end{pmatrix}, \quad \Delta\phi_l = \begin{pmatrix} \Delta\phi_{min} \\ \Delta\phi_{cn} \\ \Delta\phi_{cp} \\ \Delta\phi_e \end{pmatrix}, \quad \Delta\phi_r = \begin{pmatrix} \Delta\phi_{cn} \\ \Delta\phi_{cp} \\ \Delta\phi_e \\ \Delta\phi_{max} \end{pmatrix} \tag{56}$$

where $c_{\phi n}$, $c_{\phi c}$, $c_{\phi p}$, $c_{\phi e}$ = torsional stiffness in the negative, central, positive, and enhanced (positive second stage) phases, correspondingly; $b_{\phi n}$, $b_{\phi c}$, $b_{\phi p}$, $b_{\phi e}$ = free members respectively; $\Delta\phi_{min}$, $\Delta\phi_{max}$ = conditional limit values of the TVD relative rotation, such that $\Delta\phi_{min} < \Delta\phi < \Delta\phi_{max}$; $\Delta\phi_{cn}$, $\Delta\phi_{cp}$ = the negative and positive boundaries of the central section in the vicinity of zero strain; and $\Delta\phi_e$ = TVD's relative rotation angle value corresponding to the second step of stiffness.

Then, the elastic component derivative can be obtained as follows:

$$\frac{\partial T_{D\phi}}{\partial \Delta\phi} = \frac{\partial}{\partial \Delta\phi}\left(c_\phi\Delta\phi + b_\phi\right)^T(E_-(\Delta\phi - \Delta\phi_l) - E_-(\Delta\phi - \Delta\phi_r)) +$$
$$+ \left(c_\phi\Delta\phi + b_\phi\right)^T\frac{\partial}{\partial \Delta\phi}(E_-(\Delta\phi - \Delta\phi_l) - E_-(\Delta\phi - \Delta\phi_r)) = \tag{57}$$
$$= c_\phi^T(E_-(\Delta\phi - \Delta\phi_l) - E_-(\Delta\phi - \Delta\phi_r)) + \left(c_\phi\Delta\phi + b_\phi\right)^T(\delta(\Delta\phi - \Delta\phi_l) - \delta(\Delta\phi - \Delta\phi_r))$$

where $\delta(\Delta)$ = the impulse delta function.

Notice that by its definition the delta function makes sense only at one point, zeroing the multiplier value of the factor $c_\phi \Delta\phi + b_\phi$ in all other points, except for $\Delta\phi_l$, $\Delta\phi_r$. Therefore, the last member in Equation (57) can be omitted for numerical integration.

Considering that TVD operates in conditions of the TC working fluid, it can be assumed that the absorbing component is mainly of a viscous nature, and the pure dry friction can be neglected.

$$T_{D\omega}(\Delta\omega) = c_\omega \Delta\omega \tag{58}$$

Then, the derivative of absorbing component is:

$$\frac{\partial T_{D\omega}}{\partial \Delta\omega} = \frac{\partial}{\partial \Delta\omega}(c_\omega \Delta\omega) = c_\omega \tag{59}$$

*2.7. Tire Model (W)*

Considering that this study emphasizes powertrain dynamics modeling, the precise tire model needed for a four-wheel vehicle may be replaced with the simplest version of the Magic Formula [26] for describing the one-wheel longitudinal traction force ($R_{wx}$):

$$R_{wx} = R_{wx}(s, R_{wz}) = R_{wz}D\sin(C\arctan(Bs - E(Bs - \arctan(Bs)))) \tag{60}$$

where $R_{wz}$ = vertical reaction at the tire contact point, $B$, $C$, $D$, $E$—coefficients for dry asphalt, $s = (v_r - v_x)/|v_x|$—longitudinal slip, $r_{we}$ = wheel effective rolling radius, $v_x$ = wheel center longitudinal speed, $v_r = r_{we}\omega_w$—no slip equivalent speed, and $\omega_w$ = wheel angular speed. In turn:

$$r_{we} = r_0 + \Delta r_k - \frac{R_{wz0}}{C_{fz}}\left(D_{ref}\arctan\left(B_{ref}\Delta r_z \frac{C_{fz}}{R_{wz0}}\right) + F_{ref}\Delta r_z \frac{C_{fz}}{R_{wz0}}\right) \tag{61}$$

where $R_{wz0}$ = the wheel's normal vertical load, $D_{ref}$, $B_{ref}$, $F_{ref}$ = coefficients, $r_0$ = the wheel's free radius, $C_{fz}$ = tire initial radial stiffness, $\Delta r_z = r_0 - r_{wd} + \Delta r_k$ = radius deflection, $r_{wd} = r_0 - \Delta r_d$—the wheel's dynamic radius, $\Delta r_d$ = the wheel's dynamic radial deformation, and $\Delta r_k$ = the wheel's radius increment, depending on the angular speed causing the centrifugal forces.

In its turn, $C_{fz} = \frac{R_{wz0}}{r_0}\sqrt{q_{fz1}^2 + 4q_{fz2}^2}$, $\Delta r_k = q_{v1}r_0\left(\frac{\omega_w r_0}{v_0}\right)^2$

where $q_{fz1}$, $q_{fz2}$, $q_{v1}$—coefficients [26].

The moment of the wheel rolling resistance can be defined as:

$$M_{wy} = R_{wx}(r_{we} - r_{wd}) + R_{wz}f_r r_{wd}\arctan(v_r/v_0) \tag{62}$$

where $f_r$ = the rolling resistance coefficient, and $v_0$ = the speed at which the empirical measurements were made. In turn, $f_r = q_{sy1} + q_{sy3}|v_x| + q_{sy4}(v_x/v_0)^4$, and $\Delta r_d = R_{wz}/C_{fz}$, where $q_{sy1}$, $q_{sy3}$, $q_{sy4}$—coefficients [26].

*2.8. Vehicle Translational Motion Model*

Consider a simplified model of the vehicle translational motion shown in Figure 5. The conditional wheel moves on the surface with a variable slope $\alpha$. The *xy* coordinate system slides over the reference surface, so that the *x*-axis is coincident with the direction of the mass center speed ($v_x$).

The dynamics equation for the conditional vehicle has the following view in vector form:

$$m\vec{a} = \vec{R}_x + \vec{R}_z + m\vec{g} + \vec{F}_a \tag{63}$$

where $\vec{R}_x$ = total traction force of all the wheels, $m\vec{g}$ = gravity force, $\vec{F}_a$ = external aerodynamic drag force, and $\vec{a}$ = acceleration of mass $m$ center.

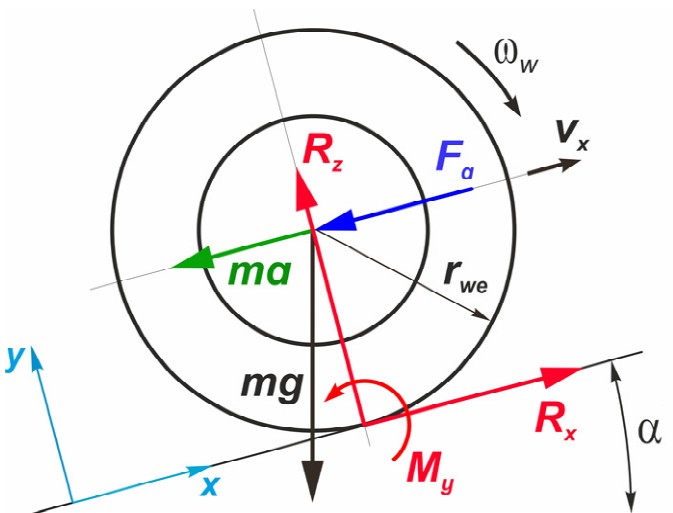

**Figure 5.** Scheme of forces acting on a conditional vehicle.

In projections onto the ***xy*** coordinate system, associated with the reference surface, Equation (52) yields:

$$\begin{cases} ma_x = R_x - mg\sin\alpha - F_a \\ R_z = mg\cos\alpha \end{cases} \tag{64}$$

where $\alpha$ = slope angle (can be a path function), and $a_x = dv_x/dt$ = translational acceleration along the road surface. The aerodynamic resistance is given by [24]:

$$F_a = \frac{1}{2}\rho_a C_x A_f v_x^2 \tag{65}$$

where $\rho_a$ = air density under the normal conditions, $C_x$ = aerodynamic drag coefficient, and $A_f$ = frontal (projective) vehicle square.

Then, for integration from Equation (53) can be derived:

$$\frac{dv_x}{dt} = \frac{R_x}{m} - g\sin\alpha - \frac{F_a}{m}, \quad \frac{dx}{dt} = v_x \tag{66}$$

where $x$ = path length, and $\alpha_s = \sin\alpha$.

### 2.9. System of Motion Equations

An automatic transmission model, based on the second-order differential tie between the engine and TC, may also be introduced for comparison and further application. Thus, the two approaches will differ by using the "conventional" and "advanced" models.

### 2.9.1. Conventional Model

The motion equations for the simple model can be easily obtained by combining the Equations (53), (54), and (64)–(66):

$$\frac{d}{dt}\begin{pmatrix} \omega_e \\ \omega_T \\ \Delta\phi \\ \psi \\ v_x \\ x \end{pmatrix} = \begin{pmatrix} (T_e - T_I)/I_e \\ (T_T - M_{tr})/I_{tr} \\ \omega_e - \omega_T \\ u_\psi \\ \frac{R_x - mg\sin\alpha - F_a}{m} \\ v_x \end{pmatrix} \tag{67}$$

In this case, as noted from the state vector, no matching exists at the level of the angular speed of the engine shaft and the differential kinematic relationship between $\omega_e$ and $\omega_T$. The $i_{tc}$ value will be calculated as a ratio consequence of the integrated values.

### 2.9.2. Advanced Model

In testing the proposed approach for modeling the hydro-mechanical transmission dynamics, the system was restricted with equations connecting the basic kinematic parameters, such as the engine shaft angular speed and vehicle translational velocity at external loading conditions and accelerator (throttle) control. Denote:

$$z = \begin{pmatrix} \varepsilon_e & \omega_e & i_{tc} & \Delta\phi & \psi & v_x & x \end{pmatrix}^T, f = \begin{pmatrix} T_T & M_{tr} & R_x & \alpha_s & F_a \end{pmatrix}^T, u = u_\psi \tag{68}$$

where $z$ = state-space vector, $f$ = vector of external factors, and $u$ = control vector.

Form templates for zero matrices: $A$ is a matrix $7 \times 7$ of system, $B$ is matrix $7 \times 1$ of control, and $C$ is matrix $7 \times 5$ of external influences. Non-zero coefficients depend on the mode of TC operation: converting or locked-up:

$$w = \frac{1}{\omega_e}, \; j = \frac{1}{I_{tr}}, \; a_{1,1} = \frac{1}{I_e}\left(\frac{\partial T_e}{\partial \omega_e} - S\right), \; a_{2,1} = 1, \; a_{3,3} = -w\varepsilon_e, \; a_{7,6} = 1,$$

$$b_1 = \frac{1}{I_e}\frac{\partial T_e}{\partial \psi}, \; b_5 = 1, \; c_{1,2} = -c_{1,1}, \; c_{3,1} = wj, \; c_{3,2} = -c_{3,1}, \; c_{6,3} = \frac{1}{m}, \; c_{6,4} = -g, \; c_{6,5} = -c_{6,3}$$

*Converting mode.* Composing Equations (25), (18), and (66), as well as considering Equations (19), (20), (44), (45), (48), and $T_T = T_I k_{tc}$:

$$Q = \frac{\lambda'_{i_{tc}}}{I_e}\frac{\partial T_I}{\partial \lambda_I}, \; S = \frac{\partial T_I}{\partial \omega_e}, \; a_{1,2} = 0, \; a_{1,3} = Qw\varepsilon_e, \; a_{4,2} = 0, \; a_{4,3} = 0, \; c_{1,1} = -Qc_{3,1}$$

*Lock-up mode.* Composing Equations (52), (18), (54), and (66), as well as considering Equations (55)–(59) and $T_T = T_D$:

$$Q = \frac{1}{I_e}\frac{\partial T_{D\phi}}{\partial \Delta\phi}, \; S = \frac{\partial T_{D\omega}}{\partial \Delta\omega}, \; a_{1,2} = -Q, \; a_{1,3} = Q\omega_e, \; a_{4,2} = 1, \; a_{4,3} = -\omega_e, \; c_{1,1} = \frac{Sj}{I_e}$$

Then, the system of equations in vector-matrix form, considering the introduced designations:

$$\dot{z} = Az + Bu + Cf \tag{69}$$

Such a format of representing equations is convenient for programming and allows perceiving the equations structurally. So, matrix $B$ is responsible for the system's internal control, and matrix $C$ is associated with the external variable conditions.

### 3. Approach Validation

The correctness and accuracy of decomposing the force parameters of powertrain units should be confirmed by the fact that, after being integrated, the derivatives provide equivalent and equal values to the moments that were themselves computed from the original dependencies. To this purpose, the verification Simulink models (Figure 6) for each unit, such as the EC engine, torque converter, and torsional damper, were composed, allowing estimating the discrepancy between the direct and integrated values of the moments. All the schemes are structurally similar. The central part is a block that describes an aggregate and calculates the moment and its derivatives. The input signal block generates parameters such as $\omega_e$, $i_{tc}$, $\Delta\phi$, according to which, the moments are decomposed. These parameters are connected directly with the aggregate's block, as well as the differentiation blocks of $\Delta u / \Delta t$, after which, they are multiplied with the corresponding partial derivatives of the moments. According to the established dependencies in Equations (5), (16), and (51), the result of summing the decomposition components is sent to the integrator. The integrated

moment value is subtracted from the directly calculated moment, and the difference is statistically processed (root mean square evaluation). As the calculations carried out during 10 s of simulation show, the differences between the direct and integrated moments' values are insignificant and primarily caused by the integration errors and phase shifts of the input signals, due to the numerical differentiation.

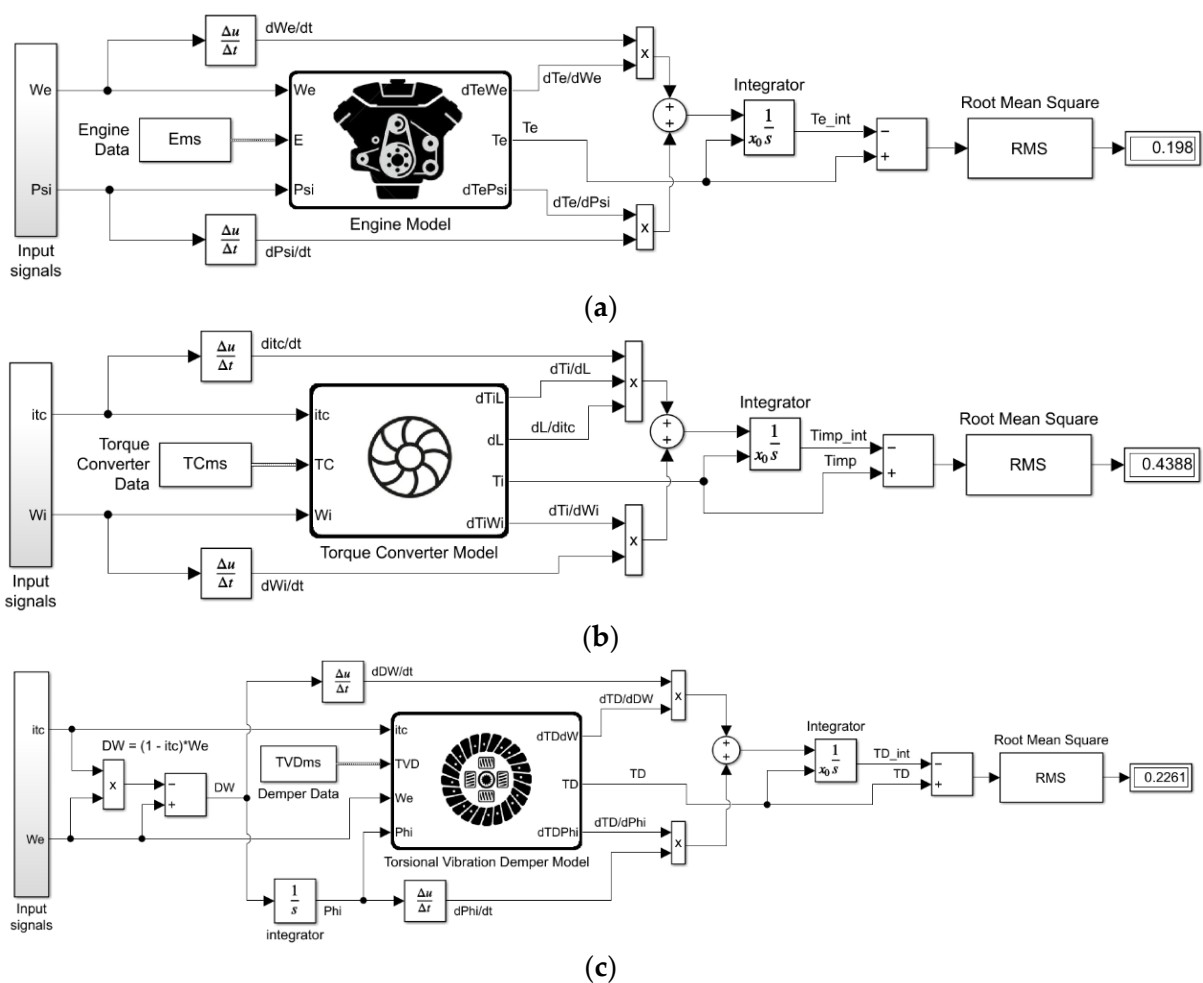

**Figure 6.** Comparing the moments' values calculated directly and by integration for: (**a**) — engine model; (**b**) —TC model; (**c**) —damper model.

## 4. Model Simulation

The section is dedicated to composing, testing, and tuning of the translational vehicle dynamics Simulink model.

### 4.1. Simulink Model Description

The Simulink model [12] of vehicle longitudinal dynamics (Figure 7) presented below is branched by subsystems, making it accessible and intuitive. The vehicle data (Table 1) for all units and mechanisms is formed as a structure array for operating as a bus to be convenient for the simultaneous transmission of all the parameters to the program functions processing the model components. The central data exchange unit is the **Bus** block that allows varying control parameters and external conditions directly in the **Vehicle Data** array at each step. Thus, the **Slope** block forms the values of the ascent angle for the entire simulation period, and the **Acc Pedal** block gives the conditionally normalized level of activating the accelerator pedal. The **Gearbox logic** block contains a gear shifting control algorithm that uses the current parameters of *States* signal, as discussed later. The model includes a **Reverse Activation** block, sending a logic signal to the gearbox control

algorithm. This option is important for validating the model workability and organizing the backward movement when studying the vehicle behavior in a limited space.

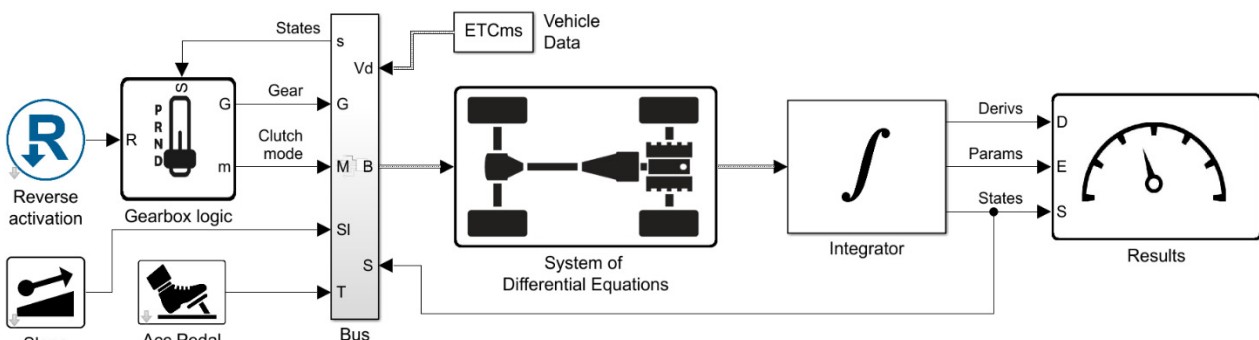

**Figure 7.** Simulink model of translational vehicle dynamics.

**Table 1.** The summary table of data accepted for the model, in the order of their appearance.

| Name | Value | Name | Value | Name | Value | Name | Value | Name | Value |
|---|---|---|---|---|---|---|---|---|---|
| $T_{ext}$ | $SS*$ | $i_{g(0)}$ | Inf | $\eta_{gG(0)}$ | 0.982 | $l_{g(0)}$ | 0.005 | $B$ | 10 |
| $k_\varphi$ | 0.65 | $i_{g(1)}$ | 4.171 | $\eta_{gG(1)}$ | 0.982 | $l_{g(1)}$ | 0.005 | $C$ | 1.9 |
| $a$ | $8.05 \times 10^{-7}$ | $i_{g(2)}$ | 2.34 | $\eta_{gG(2)}$ | 0.982 | $l_{g(2)}$ | 0.005 | $D$ | 1 |
| $b$ | $-9.2 \times 10^{-4}$ | $i_{g(3)}$ | 1.521 | $\eta_{gG(3)}$ | 0.983 | $l_{g(3)}$ | 0.004 | $E$ | 0.97 |
| $c$ | 0.0525 | $i_{g(4)}$ | 1.143 | $\eta_{gG(4)}$ | 0.983 | $l_{g(4)}$ | 0.003 | $D_{ref}$ | 9 |
| $d$ | 4.287 | $i_{g(5)}$ | 0.867 | $\eta_{gG(5)}$ | 0.985 | $l_{g(5)}$ | 0.002 | $B_{ref}$ | 0.23 |
| $\rho$ | 860 | $i_{g(6)}$ | 0.691 | $\eta_{gG(6)}$ | 0.985 | $l_{g(6)}$ | 0.001 | $F_{ref}$ | 0.1 |
| $D_a$ | 0.2762 | $i_{g(R)}$ | $-3.403$ | $\eta_{gG(R)}$ | 0.982 | $l_{g(R)}$ | 0.005 | $r_0$ | 0.327 |
| $p_0$ | 0.0031 | $I_e$ | 0.1629 | $\eta_{wB}$ | 0.995 | $c_{\phi n}$ | 621.5 | $R_{wz0}$ | 4120 |
| $p_1$ | $-6.13 \times 10^{-4}$ | $I_T$ | 0.0456 | $\eta_{dB}$ | 0.999 | $c_{\phi c}$ | 7333.9 | $q_{fz1}$ | 13.37 |
| $p_2$ | $-0.0035$ | $I_w$ | 0.4193 | $\eta_{TB}$ | 0.999 | $c_{\phi p}$ | 621.5 | $q_{fz2}$ | 14.35 |
| $p_3$ | 0.0085 | $I_d$ | 0.0490 | $\eta_{gB(0)}$ | 0.992 | $c_{\phi e}$ | 1191.8 | $q_{v1}$ | $7.1 \times 10^{-5}$ |
| $p_4$ | $-0.0178$ | $I_{g(0)}$ | 0.0846 | $\eta_{gB(1)}$ | 0.992 | $b_{\phi n}$ | $-58.576$ | $v_0$ | 16.67 |
| $p_5$ | 0.0213 | $I_{g(1)}$ | 0.1154 | $\eta_{gB(2)}$ | 0.992 | $b_{\phi c}$ | 0 | $q_{sy1}$ | 0.007 |
| $p_6$ | $-0.0113$ | $I_{g(2)}$ | 0.1077 | $\eta_{gB(3)}$ | 0.993 | $b_{\phi p}$ | 58.576 | $q_{sy3}$ | 0.0015 |
| $q_0$ | 3.6987 | $I_{g(3)}$ | 0.1 | $\eta_{gB(4)}$ | 0.993 | $b_{\phi e}$ | $-240$ | $q_{sy4}$ | $8.56 \times 10^{-5}$ |
| $q_1$ | $-8.2837$ | $I_{g(4)}$ | 0.0923 | $\eta_{gB(5)}$ | 0.995 | $\Delta\phi_{min}$ | $-0.5236$ | $m$ | 1680 |
| $q_2$ | 14.076 | $I_{g(5)}$ | 0.0846 | $\eta_{gB(6)}$ | 0.995 | $\Delta\phi_{cn}$ | $-0.0087$ | $\rho_a$ | 1.225 |
| $q_3$ | $-14.027$ | $I_{g(6)}$ | 0.0846 | $\eta_{gB(R)}$ | 0.992 | $\Delta\phi_{cp}$ | 0.0087 | $C_x$ | 0.24 |
| $q_4$ | 5.2481 | $I_{g(R)}$ | 0.1154 | $l_T$ | 0.002 | $\Delta\phi_e$ | 0.5236 | $A_f$ | 2.04 |
| $i_f$ | 3.517 | $\eta_{wG}$ | 0.985 | $l_d$ | 0.005 | $\Delta\phi_{max}$ | 0.8727 | $g$ | 9.81 |
| $i_d$ | 1 | $\eta_{dG}$ | 0.985 | $l_w$ | 0.005 | $c_\omega$ | 6.4 | - | - |

\* smoothed spline.

The system of differential equations Equations (5), (19), and (51), as well as the equations of power and kinematic interaction, are arranged in the ***System of Differential Equations*** block transmitting the states' derivatives and a set of additional necessary parameters through the output bus. In the ***Integrator*** block at each loop step, the Cauchy problem is numerically solved for the system of nonlinear differential equations. After being integrated, the *States* are routed to the corresponding central data bus to be used at the next step of calculating derivatives. The *States*, *Derivatives*, and *Parameters* are sent to the ***Results*** block for processing and logging.

## 4.2. Gearbox Control Algorithm

Figure 8 shows the *Stateflow* model (in the C language syntax) of processing the conditions and events that ensure the functioning of control algorithms for the automatic gearbox shifting and torque converter clutch's locking up (the content of the *Gearbox logic* block in Figure 4). The logic chart provides two discrete output parameters, i.e., the *gear* and *mode*, based on a minimum set of input parameters, i.e., *We*, *itc*, and *Rev*, as well as on a series of built-in threshold criteria $We_0$, *Wl*, *dWl*, *i*, *imin*, and *ilc*. The basic informational operators in the chart are conditions (*on*, *off*, *up*, *dn*, *lc*, and *un*) and events (***D***, ***N***, ***R***, ***S***, ***Up***, ***Dn***, ***Lc***, and ***Un***) that provide switching between the chart's states. The block content is being executed once it is activated (highlighted in the blue frame).

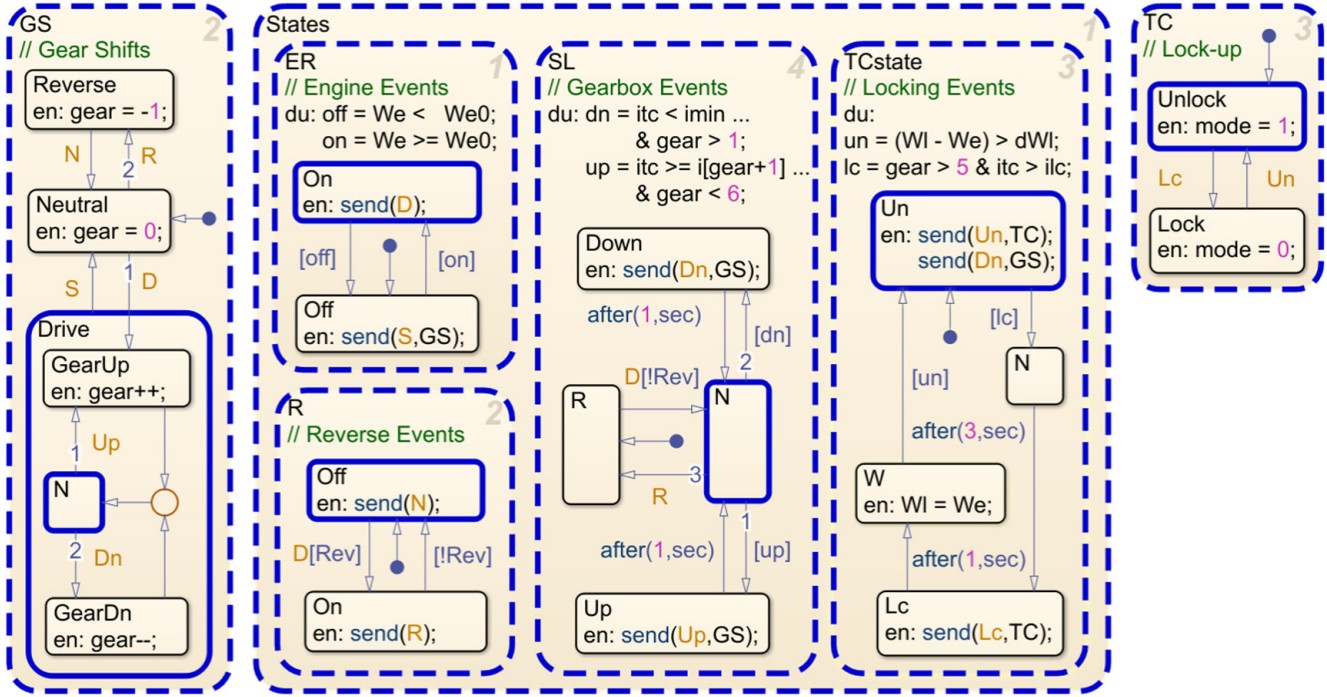

**Figure 8.** *Stateflow* chart of drivetrain control logic.

**Output variables.** The variable *gear* corresponds to the gear number and takes integer values in the range of −1, 6, where a negative value is reserved for the reverse gear, with zero for the neutral and 1–6 for the gears forward. The *mode* variable takes only one of two values (0, 1), where 1 corresponds to the torque converter's normal operating mode and 0 corresponds to the lock-up state via a clutch activation, followed by passing the torque through a torsional vibration damper.

**Input variables.** The input variable *We* tracks the current value of engine angular speed, and *itc* is the TC kinematic ratio defined by Equation (9). In practice, it can be obtained by the ratio of the signals from the angular speed sensors of the turbine and engine shafts. The *Rev* variable is boolean and only reflects the intention to engage/disengage the reverse gear.

$We_0$ limits the minimum stable idle speed and is a constant value corresponding to 250 rpm. *Wl* is a dynamic variable that memorizes a value of engine angular speed a bit later than the TC clutch locking up. *dWl* restricts the engine speed *We* reduction, relative to the recently set value *Wl*. *i* is a one-dimensional array of values setting the critical kinematic ratio for each gear. The values are not equal and determine the adjustments for the general shifting strategy. *imin* sets the lower permissible value of the TC's kinematic ratio for gears higher than first; this is one value in this algorithm. However, as in the case for *i*, an array may be specified, ensuring an individual adjustment of the conditions for downshifting. *ilc* is the critical value of the TC's kinematic ratio for the locking moment at sixth gear.

**Switching charts.** The two charts in Figure 8—*GS* and *TC*—are event-driven only.

The *TC* chart has only two states, which are controlled by the mutually exclusive events *Lc* and **Un**. By default, the torque converter is unlocked; at the start of the *Unlock* block, the conditional identifier 1 is assigned to the *mode* variable. When the *Lc* event appears, the *Lock* block is activated, and the *mode* variable is set to 0, which corresponds to activating the torque converter clutch.

The *GS* chart structurally reflects the automatic transmission gearshift scheme, which is conditionally divided into three main components: *Neutral*, *Reverse*, and *Drive*. Each state excludes the possibility of spontaneously activating the other. Block *Neutral* is the default accepted input, where the variable *gear* is set to zero, corresponding to the infinite gear ratio in the transmission model. There are two outputs from the *Neutral* block—the primary via the event **D** and secondary via the event **R**. As soon as the event **D** activates the *Drive* block, the default state switches to *GearUp*, where a gear value is increased by one only when entering. Immediately after this, an unconditional transition to Block *N* (waiting for events) is performed ($N \rightarrow GearUp \rightarrow N$). If the **Up** event appears, a circular transition throughout the *GearUp* block will be executed, followed by increasing the gear value for one more. If the **Dn** event occurs, a similar $N \rightarrow GearDn \rightarrow N$ transition is accompanied by decreasing the gear value for one. Thus, a continuous circulation of switches, separated by the **Up** and **Dn** events, is carried out. The *Drive* mode is exited to *Neutral* when the **S** event occurs. If the **R** event happens, the *Reverse* block is activated, and the value -1 is assigned to the *gear* variable, which corresponds to a negative gear ratio in the transmission model.

**Condition charts.** The charts in the *States* frame process the input data, coordinat conditions and parallel events, provide delays, and manage the main events for the *GS* and *TC* switching charts.

The *ER* chart permanently monitors the exit over the lower engine speed boundary. By default, the *Off* block is active. When *We* is greater than $We_0$, the *on* condition is assigned the logical value of 1. Therefore, the engine can be used in traction mode. When the active state transits to the *On* block, the event **D** is being sent once, allowing the activation of gear shifting. If the engine speed falls below $We_0$, the *off* condition becomes 1, passing to the block *Off* the event **S**. This instantly puts the gear selector into neutral. Such a technique facilitates preventing the stopping of calculations due to an internal error.

In the *R* chart, the *Off* block is set active by default. Suppose the input variable *Rev* is equal to 1 and **D** event has already been sent. In that case, a transition to the *On* block is carried out with the simultaneous sending the **R** event, which allows for engaging the reverse gear. With an external condition opposite to *Rev*, it switches to *Off* again, and the block sends an **N** event for transiting to neutral.

The *TCstate* chart permanently evaluates the Boolean variables *un* and *lc*. The *lc* condition is fulfilled only in 6th gear, when the ratio of the turbine and engine shaft speeds is greater than the preset *ilc*. Therefore, at the initial activation of the *Un* block (and happening the *lc* condition), the transition to the *N* block is carried out, where the state delays for 3 s. This is enough to complete the transient process after engaging sixth gear and raising the engine shaft speed. Then, the *Lc* block forwards the **Lc** event to the *TC* chart for locking the torque converter clutch.

After another 1 s, the *W* block becomes active, and the current value of the engine shaft revolutions is stored in the *Wl* variable. Then, by monitoring the condition *un* tied to decreasing the engine shaft speed (due to augmenting the load or lessening the accelerator position), a transition to the *Un* block is fulfilled. This simultaneously sends **Un** and **Dn** for unlocking the torque converter clutch and reducing the automatic transmission gear.

The *SL* chart monitors the *dn* and *up* conditions. Thus, the increase in gear is possible only if the ratio *itc* exceeds the value preset for the next gear, and the *gear* value itself does not exceed six. The condition *dn* is satisfied if the relative turbine speed drops to the critical value *imin* and the *gear* is second and higher. By default, the *R* block is activated first to exclude the **Up**/**Dn** events if an intent to engage reverse gear is given. As soon as the event **D** is sent and *Rev* corresponds to zero, the transition $R \rightarrow N$ to the event distribution state

is carried out. If the condition *up* is satisfied, the loop $N \to Up \to N$ is executed, sending the ***Up*** event and a delay for a second. Similarly, for the event ***Dn***, under the condition *dn*, the loop $N \to Down \to N$.

**The order of executing parallel charts.** The priority of chart execution significantly affects the occurrence order of control events. Thus, the *States* chart is executed first, followed by the *GS* and *TC* switching charts. Within the *States* chart, the hierarchy is also essential, following the order in which the aggregates are managed. Thus, the *ER* chart is executed first, as it is responsible for the sufficient engine speed to start moving. The second is the *R* chart, which is responsible for the reverse. The third is the *TCstate* chart, which is responsible for locking the TC's clutch. The last one is executed by the *SL* chart, which is responsible for the *up/down* of the forward gears. In this order, there are no contradictions between the charts, and the general algorithm works stably.

*4.3. Model Verification*

**Full throttle.** The simplest way for verifying the model may be arranged by comparing the external and simulated data for the vehicle's maximum performance mode. There is a set of values for vehicle translational speeds at the corresponding time points (Table 2) that can be used as criteria for this mode. Setting the position of the accelerator equal to 1 and slope equal to 0, the output characteristics (depicted in Figure 9) can be obtained.

**Table 2.** Comparison of acceleration data by source [22] and by the model simulation.

| Speed, km/h | 60 | 100 | 120 | 130 | 150 | 180 | 200 | 210 | 220 |
|---|---|---|---|---|---|---|---|---|---|
| Source data, s | 3.1 | 6.3 | 9.2 | 10.6 | 13.6 | 21 | 27.5 | 32.3 | 39.3 |
| Simulation data, s | 3.15 | 6.34 | 8.7 | 9.94 | 13.1 | 19.8 | 26.9 | 31.3 | 39 |
| Error, % | −1.61 | −0.63 | 5.43 | 6.22 | 3.68 | 5.71 | 2.18 | 3.1 | 0.76 |

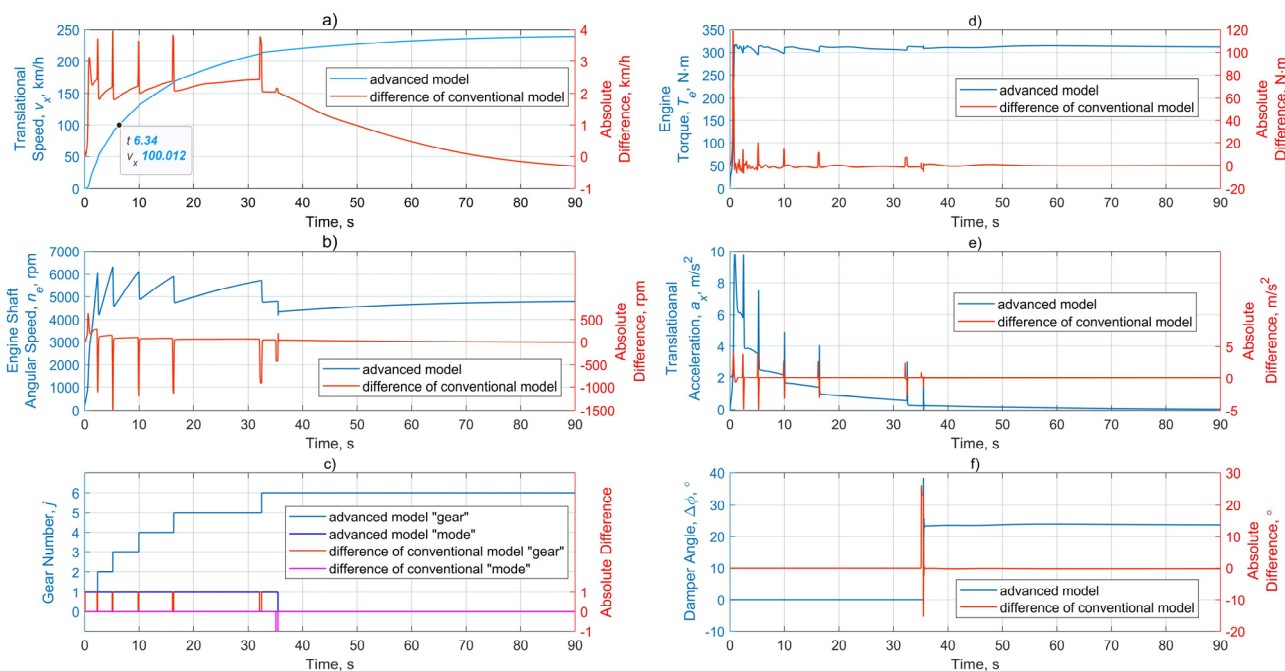

**Figure 9.** Performance parameters for two model variants with full fuel supply: (**a**)—speed, (**b**)—engine rpm, (**c**)—gear shifting and TC's clutch locking up, (**d**)—engine torque, (**e**)—acceleration, and (**f**)—damper deformation angle.

The primarily represented calculation data are the results of the advanced model, and the simulation results for the conventional model are given as an absolute difference,

relative to the advanced model, in order to avoid overlapping graphs. As it can be seen, the difference in the outputs of the models' maximum performance is mainly caused by a phase bias (i.e., some delay in the advanced model, relative to the conventional one). However, the advanced model's results fit better than those given by the source [22], and they provide insignificant calculation errors.

Some inconsistency of the data in Table 1 can also be explained by the fact that, in the source [22], the 3.2 FSI engine's torque curve, which was synthesized on the manufacturer's data with a maximum value of 330 Nm, was used. In contrast, this study's engine torque curve, with a peak value of 317 Nm, is based on the accurate characteristic provided by the bench tests [23]. In general, it can be concluded that the proposed model fully reflects the physical processes when accelerating to a maximum speed.

Note that the peak surges inevitably appear when gear shifting, due to the absolute rigidity of transmission elements' and tire's models. This effect is caused by instantly changing the ratio and inertia moments in the transmission. These impulses in the actual conditions are mitigated with the drivetrain parts' flexibility during the automatic gearbox transient processes and are due to the tire tangential elasticity.

**Backward movement.** First, the reverse may be an additional test means, demonstrating the model consistency, in general, as well as the efficiency of the gearbox control algorithm. To this purpose, a zero road's slope and trapezoidal accelerator control law (with a slight upper level) were set. The results shown in Figure 10 were obtained after performing a simulation within 10 s for both model versions.

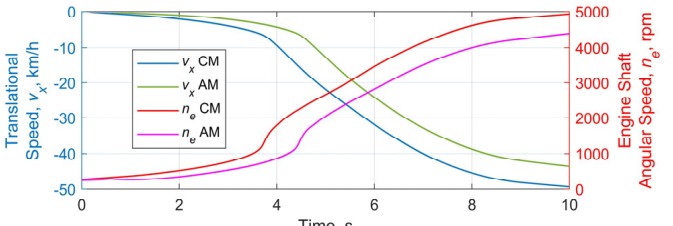 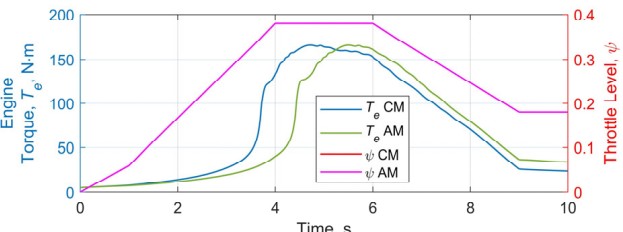

**Figure 10.** The main models' parameters for comparing at reverse driving.

As depicted in Figure 10, despite the similarity of the speed and engine revolution curves, the main discrepancies in kinematics are caused by the relative lag of the advanced model (AM) torque, compared to the conventional model (CM). Such a picture shows that the conventional model's sensitivity at low powers is higher than that of the advanced one, which is caused, to a greater extent, by the transient processes.

### 4.4. Virtual Tests

Several virtual tests may be performed, providing the same control and load signals for two model types within 100 s of simulation.

### 4.4.1. Fixated Accelerator Level and Variable Slope

Under such initial conditions, when an external load is variable and accelerator position is unchanged, the vehicle transmission should automatically adapt to the driving circumstances, mainly owing to the TC turbine sliding and gear shifting. The left upper part in Figure 11 shows the nature of changing the ascent angle in two stages, with a maximum value of 30 degrees. At the same time, the increases and decreases in the grade and sections of the constant slope are equally represented. As seen in the figure's lower left picture, the engine has enough power to augment the vehicle speed, even while climbing up to 40th s and upshifting gears from first to fifth. From the 40th s, due to a series of downshifting up to the second gear, the load changes with a long slope rising, causing a significant decrease in turbine speed. Starting from the 60th s, the grade intensity decreases, external resistance reduces, and engine rises speed within 15 s, followed by upshifting to fifth gear by 87th

s. At the same time, as can be seen in the lower right picture, the TC efficiency is strictly within the range with the lower limit of 0.8.

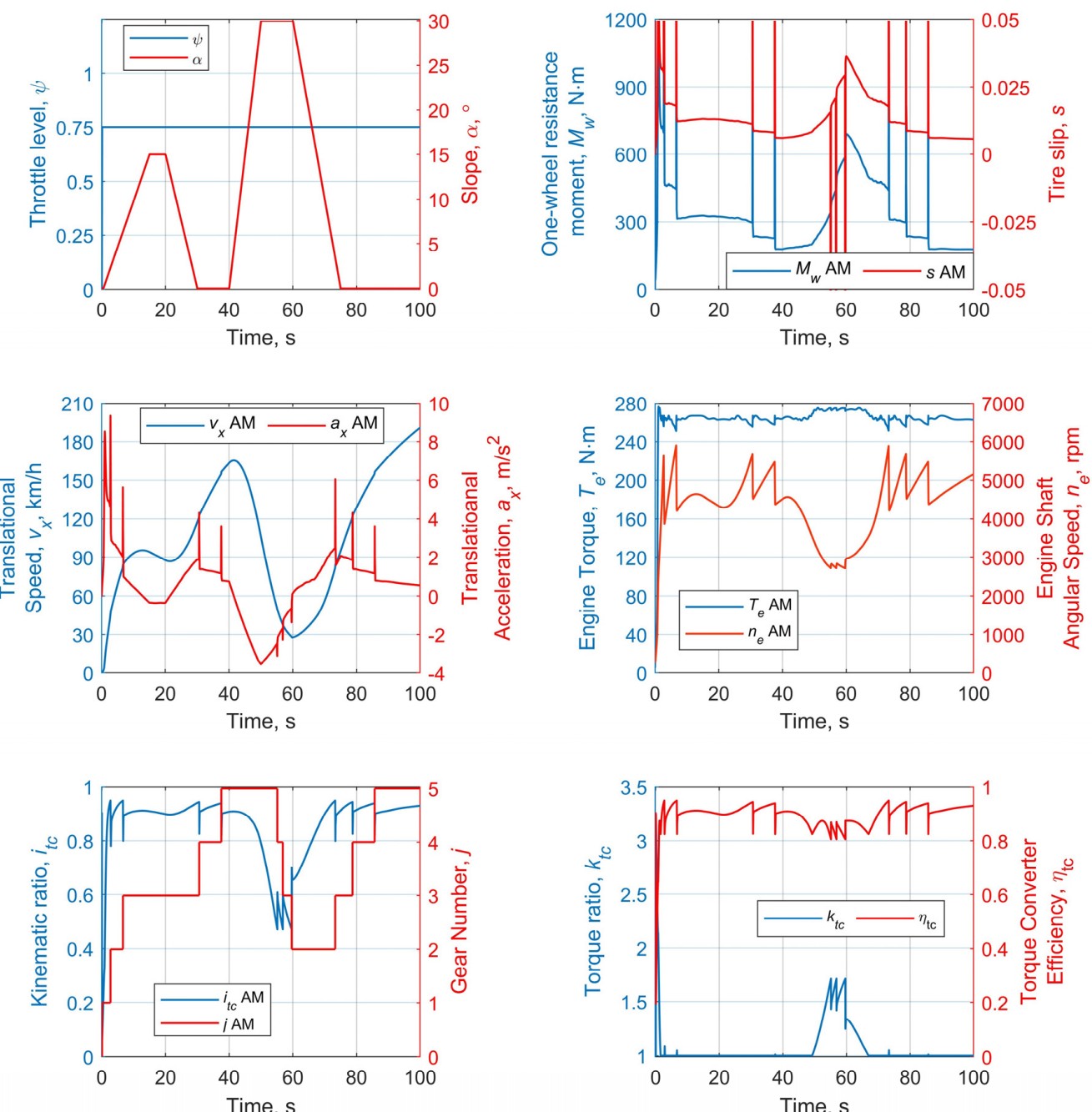

**Figure 11.** Simulation results for variable slope at accelerator level 0.75.

Note that the torque level remains approximately constant and fits the established fuel supply pattern. The engine uses a range of 3000–6000 rpm, which corresponds, on the one hand, to the operation by engine's partial power characteristic and, on the other hand, to the frequency offset to a range of higher torque. Note that both the conventional (CM) and advanced (AM) models show a high coherence, with minor differences in this test; therefore, the results are presented only for the AM variant. This test's primary purpose is to demonstrate the stability of the proposed control algorithm, in general, when the automatic drivetrain adaptation can be provided. The gearbox logic may both down- and upshift, quickly returning to a state of higher gears and speeds without throttle manipulation.

### 4.4.2. Variable Throttle Level and Variable Slope

Another option for checking the stability of automatic control consists of reducing the fuel supply with increasing external resistance. During the first 10 s of acceleration, the throttle level linearly reaches the maximum position (Figure 12). It is held for some time; then, it decreases to the half level between 30th and 45th s, after which the position is unchanged until test completion. In this case, the external load, in the form of slope, increases stepwise, up to 30 degrees, and remains constant for the last 25 s.

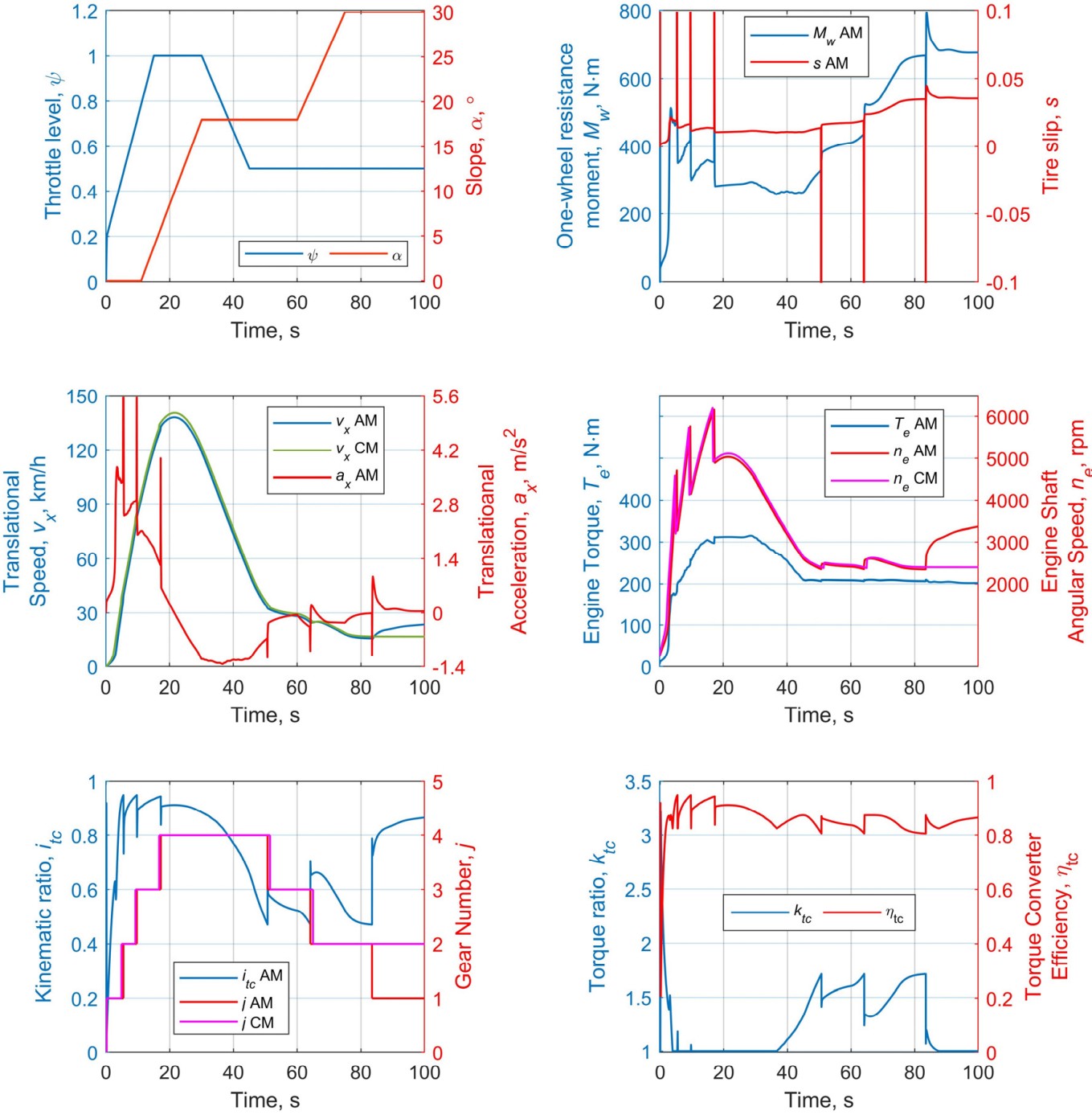

**Figure 12.** Simulation results for simultaneous variable accelerator level and variable load.

The speed change nature entirely agrees with the combined factors of the control and load. The outputs of both models are identical in tendency, but with some differences.

Thus, the fuel supply decrease from the 30th s, followed by the slope angle increase, leads to an intensive downshifting up to the 1st gear in the AM case and 2nd gear in the CM case. At the same time, the AM continued to accelerate slightly, while CM stabilized its speed. Note that, between the 35th and 85th s, the torque converter is intensively included in the work, with an augmenting torque ratio up to 1.7, which reduces gears and provides a significant growth of the total transmission ratio. It is additionally worth noting that the engine torque shape matches the form of the accelerator control law and speed change tendency, which also indicates the control stability, depending on the driving mode.

## 5. Conclusions

In the framework of the research plans, the main goal of this study concerns the methodical and math bases: all the fundamental stages of modeling a vehicle with an automatic powertrain are systematized and worked out; the mathematical apparatus is widely represented; the simple and effective automatic control algorithm is proposed, and its implementation as the *Stateflow*-chart is delivered; the original approach was proposed for describing the interaction model of hydro-mechanical transmission units; two variants of the powertrain dynamics model were composed and tested; and the described models' operability and efficiency were confirmed.

Both models, conventional and advanced, provide resembling results, which increases the adequacy of the likelihood of the newly proposed technique, which is essential for its further development and use. The main advantage of the proposed approach is the stricter consideration of the transient processes that are coordinated at the higher order derivative, namely the engine shaft's angular acceleration, which can be further used as a state parameter to be minimized in optimal control search. Additionally, the proposed model form is derived in the classical state-space view, which makes it convenient for linearization, as well as use in MPC optimization schemes.

As seen, according to the results of testing the gearbox logical control algorithm, there is no need, in general, for coordinating the gear shifts with vehicle speed values. In this study, only the single critical value of the lower threshold is used, which, in some cases of powertrain operation, leads to a significant decrease in engine speed, without downshifts. However, this significantly augments the turbine torque, without overswitching. Moreover, it is logical to assume an individual lower threshold for each gear. Different options for tuning the transmission are possible for getting more advantages from both the mechanical and hydrodynamic transmission parts. Such settings should be based on the balance of sufficient traction and vehicle fuel efficiency. The last point determines the optimal control problem, when the fixed gear shifting algorithm will be opposed to the control, based on online optimization procedures for a prediction horizon.

The proposed approach can be used for conducting further research on modeling the traction control based on the MPC. The model is proven to be stable and provides high-speed calculations, which is essential for its use in SIL-HIL modeling cases. Thus, this model will serve as a basis for creating more complicated powertrain models, elaborating the advanced control algorithms, and supporting the means for testing and comparing the automated and optimal control (concerning their efficiency).

**Author Contributions:** Conceptualization, M.D. and S.M.E.; methodology, M.D.; software, M.D.; validation, M.D. and S.M.E.; formal analysis, S.M.E.; investigation, M.D.; resources, S.M.E.; data curation, M.D.; writing—original draft preparation, M.D.; writing—review and editing, S.M.E.; visualization, M.D.; supervision, S.M.E.; project administration, S.M.E.; funding acquisition, S.M.E. All authors have read and agreed to the published version of the manuscript.

**Funding:** This research is financially supported by the Natural Sciences and Engineering Research Council of Canada (grant no. RGPIN-2020-04667).

**Institutional Review Board Statement:** No institutional review is required.

**Informed Consent Statement:** Not applicable.

**Data Availability Statement:** Some (or all) data, models, or code that support the findings of this study are available from the corresponding author upon reasonable request.

**Acknowledgments:** The authors are grateful to three anonymous reviewers for their thorough and most helpful comments.

**Conflicts of Interest:** The authors declare no conflict of interest.

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
