# Peer review of "Modeling Combined Operation of Engine and Torque Converter for Improved Vehicle Powertrain’s Complex Control"

_vehicles, doi:10.3390/vehicles4020030_

Round 1

Reviewer 1 Report

The manuscript is written in an exhaustive manner. It sounds more like a report than an article. 

The paper needs to be extensively re-formatted. I will recommend authors try to be more concise about the subject topic.

It's hard to grasp the research idea, problem statement, and research methodology. 

The introduction part is quite bulky. There are no references to the statements that the author has made in the introduction section such as "In modern vehicles with automatic transmission, various algorithms are used to control the powertrain logic based on processing the data from several sensors and providing variable operating modes (sport, economy, etc.)." its better to cite related references, throughout the manuscript.

Normally literature review is an integral part of the introduction unless it is necessary to discuss something in detail.

The reference citations are not as per the journal standards. At one instance it is [1] at another it is (Mahmoud et al). Quite irregular.

Sections 4 and 5 can be one as results and validation. 

Author Response

Thank you. Please, see attached file.

Reviewer 2 Report

The paper is very well researched, the presented approach is very useful for future work. The authors clearly presents the simulation approach and the developed models.  The results are in line with the proposed modeling and simulation. The conclusions are clearly following the achievements of he paper.   

Author Response

Thanks a lot. We've corrected English where possible.

Reviewer 3 Report

The authors continue the tradition of describing the car as a control object. At the same time, without taking into account the physical processes taking place in the engine, drive train and wheel. The scientific novelty of the work as formulated by the authors is not obvious. The results of minimizing fuel consumption and other car indicators are known, for example in the Amesim software package, which uses a simulation of dynamical systems based on Bond Graf. Including optimization of the gear selection algorithm. The proposed model contains a large amount of empiricism (coefficients, characteristics). How the authors intend to obtain these values.

Author Response

Thanks a lot. See, please, the attached file.

Round 2

Reviewer 1 Report

Thanks for the reply